# Insights into phosphoethanolamine cellulose synthesis and secretion across the Gram-negative cell envelope

Preeti Verma ®[1], Ruoya Ho[1], Schuyler A. Chambers ®[2], Lynette Cegelski ®[2] ✉ & Jochen Zimmer ®[1,3] ✉

Phosphoethanolamine (pEtN) cellulose is a naturally occurring modified cellulose produced by several Enterobacteriaceae. The minimal components of the *E. coli* cellulose synthase complex include the catalytically active BcsA enzyme, a hexameric semicircle of the periplasmic BcsB protein, and the outer membrane (OM)-integrated BcsC subunit containing periplasmic tetratricopeptide repeats (TPR). Additional subunits include BcsG, a membrane-anchored periplasmic pEtN transferase associated with BcsA, and BcsZ, a periplasmic cellulase of unknown biological function. While cellulose synthesis and translocation by BcsA are well described, little is known about its pEtN modification and translocation across the cell envelope. We show that the N-terminal cytosolic domain of BcsA positions three BcsG copies near the nascent cellulose polymer. Further, the semicircle's terminal BcsB subunit tethers the N-terminus of a single BcsC protein in a trans-envelope secretion system. BcsC's TPR motifs bind a putative cello-oligosaccharide near the entrance to its OM pore. Additionally, we show that only the hydrolytic activity of BcsZ but not the subunit itself is necessary for cellulose secretion, suggesting a secretion mechanism based on enzymatic removal of translocation incompetent cellulose. Lastly, protein engineering introduces cellulose pEtN modification in orthogonal cellulose biosynthetic systems. These findings advance our understanding of pEtN cellulose modification and secretion.

Bacterial biofilms have major impacts on our healthcare, industrial, and infrastructure systems. In a biofilm community, bacteria are encased in and protected by a complex biopolymer meshwork including protein polymers, polysaccharides, and nucleic acids[1,2]. Cellulose is a common bacterial biofilm component produced by a variety of prokaryotes, such as *Escherichia coli* (*Ec*), *Salmonella enterica*, and *Gluconacetobacter/Komagataeibacter xylinus* (*Gx*)[3]. These Gram-negative bacteria synthesize cellulose and secrete it across the cell envelope via the Bacterial cellulose synthase (Bcs) complex. The Bcs complex consists of components located in the inner and the outer

membrane (IM and OM, respectively), as well as the periplasm[4]. Its core machinery contains the catalytically active cellulose synthase BcsA enzyme that forms a functional complex with the periplasmic but IM-anchored BcsB subunit[5,6]. A third component, BcsC, likely creates a cellulose channel in the OM[7,8]. In the periplasm, the conserved cellulase BcsZ has been shown to be important for cellulose production in vivo[9–12].

BcsA is a processive family-2 glycosyltransferase (GT) that synthesizes cellulose, a linear glucose polymer, from UDP-activated glucosyl units. Additionally, the enzyme also secretes the nascent

[1]Department of Molecular Physiology and Biological Physics, University of Virginia School of Medicine, Charlottesville, VA, USA. [2]Department of Chemistry, Stanford University, Stanford, CA, USA. [3]Howard Hughes Medical Institute, Chevy Chase, MD, USA. ✉e-mail: cegelski@stanford.edu; jochen_zimmer@virginia.edu

cellulose chain across the IM through a channel formed by its own transmembrane (TM) segment[5]. Thereby, cellulose elongation is directly coupled to membrane translocation. The periplasmic BcsB subunit is required for BcsA's catalytic activity[6]. It contains two structural repeats consisting of a region resembling a carbohydrate-binding domain (CBD) that is fused to a flavodoxin-like domain (FD)[5]. Lastly, the C-terminal segment of BcsC forms a 16-stranded β-barrel in the OM[8]. This pore is preceded by 19 predicted periplasmic tetratricopeptide repeats (TPR), which consist of a pair of anti-parallel α-helices connected by a short loop. BcsC's TPRs likely form an α-helical solenoid structure bridging the periplasm[13].

*E. coli* and other Enterobacteriaceae can modify cellulose with lipid-derived phosphoethanolamine (pEtN) at the sugar's C6 position[1,14]. This modification promotes biofilm cohesion, is required for hallmark biofilm macrocolony wrinkling phenotypes, and enhances cell association of curli amyloid fibers and adhesion to host tissues[14–17]. Transfer of pEtN is catalyzed by the membrane-bound pEtN transferase BcsG[15], Fig. 1a. Recent cryogenic electron microscopy (cryo-EM) studies of the *Ec* Bcs complex provided the first insights into the organization of the pEtN cellulose biosynthesis complex, Fig. 1b, c[18–20]. Importantly, a single BcsA subunit associates with six BcsB copies that form a semicircle at the periplasmic water-lipid interface. BcsA sits at one end of the semicircle where it steers the secreted cellulose polymer towards the circle's center. The open half of the semicircle has been proposed to accommodate multiple BcsG copies[20].

Here, we have addressed important aspects of pEtN cellulose formation and translocation across the periplasm and the OM. We demonstrate that BcsA recruits three copies of BcsG via its N- and C-terminal domains. These domains are sufficient for BcsG binding, allowing the introduction of the pEtN modification in orthologous cellulose biosynthetic systems, wherein BcsG modifies cellulose using pEtN groups derived from lipids. Further, we show that the OM BcsC subunit binds the sixth subunit of the BcsB semicircle via its extreme N-terminus to establish an envelope-spanning Bcs complex. The cryo-EM structure of BcsC reveals that its periplasmic domain is flexibly attached to the β-barrel. Further, BcsC binds a putative cello-oligosaccharide at its TPR solenoid, likely to facilitate translocation across the periplasm. Lastly, in vivo cellulose secretion assays reveal that the cellulase activity of BcsZ is critical for cellulose secretion. While cellulose export is dramatically reduced in the absence of BcsZ, the conserved enzyme can be functionally replaced with an off-the-shelf cellulase. Our data suggest periplasmic translocation of cellulose along the BcsC solenoid and trimming of translocation incompetent cellulose by BcsZ to restore translocation to the cell surface.

## Results

The *Ec* pEtN cellulose biosynthesis machinery catalyzes the synthesis, secretion, and pEtN modification of cellulose (Fig. 1a, b). The mechanism of cellulose synthesis and translocation across the IM has previously been addressed using the BcsA and BcsB components from *Rhodobacter sphaeroides* (*Rs*), producing unmodified cellulose[21,22].

### BcsA associates with three copies of the pEtN transferase BcsG

BcsG is a membrane-bound pEtN transferase containing five N-terminal TM helices, followed by a periplasmic catalytic domain, Figs. 1d and S1a[23–25]. The homologous enzymes EptA and EptC catalyze pEtN modification of lipid A and N-glycans, respectively[26,27]. Mechanistically, BcsG likely employs a catalytic triad involving Ser, His, and Glu residues. The conserved Ser residue (Ser278) is assumed to serve as the nucleophile to attack the electrophilic phosphorous of a phosphatidylethanolamine (PE) lipid to form a covalent reaction intermediate with pEtN and releasing diacylglycerol, Figs. 1a and S1b[25]. The pEtN group is then subject to attack by the glucose C6 hydroxyl oxygen, resulting in transfer of the pEtN group to cellulose and release of the phospho-enzyme intermediate. This transfer reaction requires the

reorientation of BcsG's catalytic pocket away from the membrane towards the translocating cellulose polymer. Approximately half of cellulose's glucosyl units are modified by pEtN in *Ec* and *Salmonella* species under normal growth conditions[14].

Previous intermediate-resolution cryo-EM analyses of the *Ec* Bcs macro-complex suggested the presence of two BcsG subunits associated with BcsA[20]. Although only the TM regions were resolved, the subunits were located 'in front' of BcsA near the periplasmic exit of its cellulose translocation channel. Taking advantage of recent improvements in 3-dimensional variability analyses and classifications implemented in the cryoSPARC data processing workflows[28], we reprocessed the previously published cryo-EM data (EMD-23267)[20] focusing on the BcsA-BcsG complex, Fig. S2. Although still lacking well-resolved periplasmic domains, this analysis generated improved maps for BcsA together with a trimeric complex of BcsG at ~5.7 Å resolution, Figs. 1e and S2, S3a, and Table S1.

Each BcsG subunit contains five TM and two periplasmic interface helices that connect TM helices 3 and 4, Figs. 1e, f, and S1a. The TM and interface helices form a ring-shaped periplasmic corral with an opening towards the phospholipid head groups. An AlphaFold2 prediction of full-length BcsG places its periplasmic domain on top of the corral, with the catalytic Ser278 facing the membrane beneath, Figs. 1d and S1a. The predicted BcsG structure suggests an arched acidic tunnel from the membrane surface, through the corral, and towards Ser278 that could position the PE headgroup for nucleophilic attack, Fig. 1d.

In the Bcs complex, the three BcsG subunits are arranged along a slightly curved line, Fig. 1e, f. Interprotomer contacts within the BcsG trimer are mediated by TM helix 4 of one subunit and TM helices 2 and 5 of a neighboring subunit, suggesting that BcsG is prone to self-polymerize. Except for BcsA, no additional density is observed within the TM region that could correspond to other Bcs subunits, such as the single spanning subunit BcsF that has been shown to interact with BcsG and BcsE in vivo[14,18]. Indeed, as reported in a recent bioRxiv manuscript, a dimer of BcsF interacts with the N-terminal and membrane-associated domains of a BcsE dimer at a position roughly across from BcsA in the *Ec* Bcs macro-complex[29].

### BcsA's N-terminus recruits the BcsG trimer

Compared to the homologous enzymes from *Rs* and *Gx*, BcsAs encoded by pEtN cellulose producers contain about 140 additional N-terminal residues of unknown function (referred to as the NTD) as well as a diverging C-terminus, Fig. S4a. To test whether the NTD mediates interactions with BcsG, we performed AlphaFold2[30,31] complex predictions using either the full-length BcsA sequence or its NTD alone, together with three copies of BcsG's TM region. The predictions reproducibly suggest the coordination of a BcsG trimer via BcsA's NTD, Figs. 1f and S4b, which is predicted to fold into five short helices (α1-α5). In addition, helices α4 and α5 are likely stabilized by BcsA's extreme C-terminus, Fig. 1f.

Our refined cryo-EM map experimentally validates the AlphaFold2-generated BcsA-BcsG$_3$ model, Fig. 1e, f. The model was docked into the experimental map based on the location of BcsA. Its NTD as well as the three BcsG subunits were fit into the corresponding densities as rigid bodies to account for a different angle of the NTD-BcsG trimer relative to BcsA, Fig. S4c.

All BcsA NTD helices proposed to interact with the BcsG trimer are resolved in this map. Further, BcsA's C-terminal helix is observed in contact with the NTD's α4 and α5 helices, as predicted. In contrast to an earlier model that assumed a segment of the NTD to form a proper TM helix[20], the NTD resides on the cytosolic water-lipid interface with all of its helices running approximately parallel to the membrane surface, Fig. 1e.

Although only resolved at the backbone level in the cryo-EM map, the BcsG subunits interact with the NTD via their cytoplasmic loops

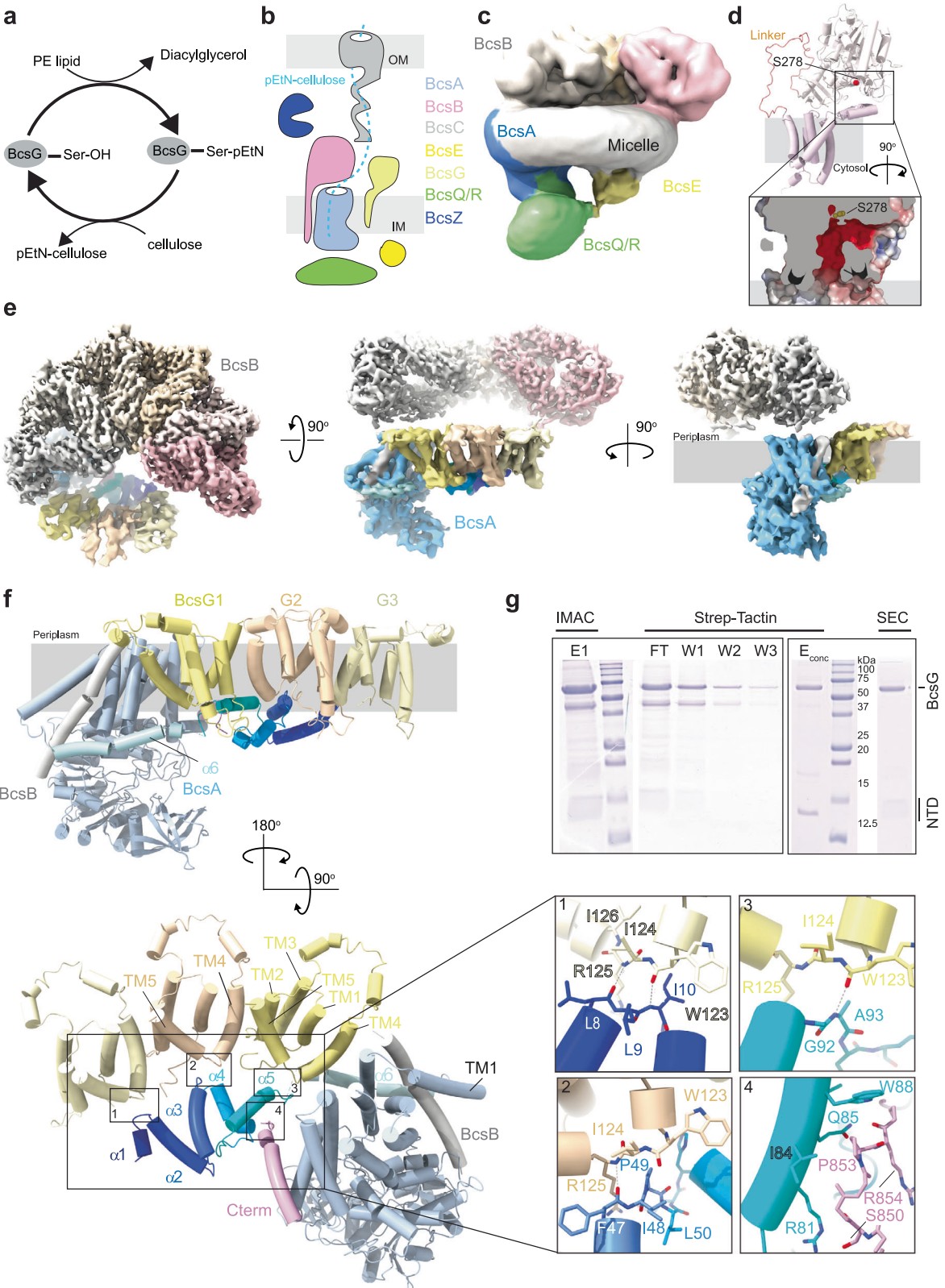

**Fig. 1 | BcsA coordinates a BcsG trimer. a** Schematic representation of the pEtN transfer reaction catalyzed by BcsG. PE lipid Phosphatidylethanolamine, Ser Ser278 as catalytic nucleophile, pEtN Phosphoethanolamine. **b** Cartoon illustration of the *Ec* cellulose synthase complex. The putative cellulose secretion path is shown as a dashed line. **c** Low-resolution cryo-EM map of the *Ec* inner membrane-associated cellulose synthase complex (IMC, EMD-23267). **d** AlphaFold2-predicted structure of *Ec* BcsG (AF-P37659-F1) with close-up showing the acidic cavity extending from the putative membrane surface. **e** Cryo-EM composite map of the IMC after focused refinements of the periplasmic BcsB hexamer and the BcsG trimer associated with BcsA, respectively. BcsB subunits are colored from light gray to pink, BcsA and trimeric BcsG are colored in shades of blue and yellow respectively, and the associated BcsB transmembrane (TM) helix is colored gray. Contour level: 5σ. **f** AlphaFold2-predicted complex of BcsA, BcsB's TM anchor, and the trimeric BcsG colored as in (**e**). **g** Co-purification of *Ec* BcsG (His-tagged) with the N-terminal domain of *Ec* BcsA (NTD, Strep-tagged) by immobilized metal (IMAC) and Strep-Tactin affinity chromatography followed by size exclusion chromatography (SEC). Source data are provided as a Source Data File.

connecting TM helices 4 and 5 (TM4/5-loop, residues 123–125), Fig. 1f. Starting with the BcsG protomer farthest away from BcsA, AlphaFold2 predicts that the TM4/5-loop forms backbone interactions with NTD residues 8-10 connecting its α1 and α2 helices. Similarly, the TM4/5-loop of the next BcsG subunit interacts with residues 47-50 between NTD's α3 and α4 helices, and the protomer closest to BcsA contacts NTD residues 92–93 following α5 via its TM4/5-loop. Past this interface, the NTD is connected to a predicted amphipathic interface helix (α6) that is only partially resolved in the cryo-EM map, Fig. 1e, f. This helix leads into BcsA's first TM helix. No direct interactions are observed or predicted between BcsG and BcsA TM segments. The BcsG TM helices fit seamlessly into the opening of the BcsB semicircle, between the first and sixth subunit, Fig. 1e.

The C-terminal helical region of *Ec* BcsA runs roughly in the opposite direction as the preceding amphipathic helix, which is present in BcsA from *Rs* and other species. This segment is resolved in our cryo-EM map and interacts with helix α5 of the NTD via an SxxPRxP motif (residues 850–856), Figs. 1e, f and S4a. The motif is predicted to interact with residues 81 through 88 (α5) of the NTD, with possible hydrogen bonds between Arg81 and Ser850 as well as Gln85 and Arg854, Fig. 1f. A similar interaction between BcsA's NTD and a BcsG trimer was recently reported on bioRxiv[29].

### BcsA's NTD is sufficient for BcsG recruitment

To test whether BcsA's NTD is sufficient to recruit BcsG, we co-expressed the Strep-tagged NTD with poly-histidine-tagged BcsG. A tandem affinity purification using Ni-NTA resin followed by Strep-Tactin beads and size exclusion chromatography indeed isolated an NTD-BcsG complex, Fig. 1g. The identities of the co-purified protein components were confirmed by Western blotting and tandem mass spectrometry sequencing, Fig. S5a–c.

Except for the NTD and the C-terminal region, *Rs* BcsA is structurally homologous to *Ec* BcsA, Fig. S5d–f. *Rs* does not encode a BcsG homolog or the other pEtN-cellulose specific Bcs components BcsE and BcsF. We sought to evaluate whether the pEtN cellulose modification could be introduced in the *Rs* cellulose biosynthetic system via BcsA engineering and inclusion of the pEtN transferase BcsG. To this end, a 'BcsA chimera' was generated containing the *Rs* BcsA sequence N-terminally extended with the *Ec* NTD (residues 1–149). In addition, the C-terminal *Rs* BcsA region (residues 729 to 788) was replaced with the corresponding *Ec* BcsA sequence (residues 828 to 872), followed by a poly-histidine tag for purification, Figs. 2a and S4a, S5f (see Methods).

The BcsA chimera was co-expressed with *Rs* BcsB as well as *Ec* BcsG and BcsF. Metal affinity and size exclusion chromatography purification of the expressed complex demonstrated the co-purification of the BcsA chimera with BcsB (the native binding partner of *Rs* BcsA) as well as *Ec* BcsG, Fig. 2b. The presence of co-purified BcsA and BcsG was confirmed by Western blotting, Fig. S5g, while BcsF was not detected in the purified sample. BcsG did not co-purify with the wild type (WT) *Rs* BcsAB complex, suggesting that the interaction is indeed mediated by the introduced *Ec* BcsA NTD, Fig. S5h.

The purified chimeric BcsAB-BcsG complex is catalytically active in vitro. Similar to previous reports on the wild type *Rs* BcsAB complex[6], the BcsA chimera synthesizes cellulose in vitro from UDP-glucose in a cyclic-di-GMP (ci-di-GMP) dependent reaction, Fig. 2c. The obtained product is readily degraded by a cellulase, as expected for a cellulose substrate.

Cryo-EM analysis of the purified complex revealed the association of the BcsA chimera with BcsG in a curved micelle, Figs. 2d, e and S6 and Table S1. Refinements of either the chimeric BcsAB complex alone or in association with BcsG resulted in maps of ~4.6 and 6 Å resolution, respectively, Figs. 2e and S3b, c, S6. The maps resolve the BcsAB complex associated with a nascent cellulose polymer in a conformation similar to the previously reported crystal structure[5], as well as the

TM domains of three BcsG subunits, Fig. 2e. The BcsG trimer is assembled as observed in the *Ec* Bcs complex described above, Fig. 1e, and interacts with short interface helices corresponding to the engineered *Ec* BcsA NTD. The also introduced *Ec* C-terminal extension is flexible and insufficiently resolved, likely resulting in the tilting of the BcsG trimer relative to BcsAB in the detergent micelle (Fig. 2d, e).

To test whether the engineered chimeric cellulose synthase complex produces pEtN cellulose in vivo, we isolated cellulosic material from the periplasm of the expression host (see Methods) where it accumulates in the absence of the OM subunit BcsC (this subunit has not been identified in *Rs* yet). The production of pEtN cellulose by the chimeric complex was validated through direct detection of material isolated from cell lysates using ¹³C cross-polarization magic-angle spinning (CPMAS) solid-state NMR spectroscopy. Compared to a reference sample of pure pEtN cellulose from *Ec*[14] and unmodified cellulose, the cellulosic material produced by the BcsA chimera in presence of BcsG is highly enriched in pEtN cellulose, Figs. 2f and S7a. Isolation of unmodified cellulose for comparison was facilitated by addition of Congo red (CR). The co-purified CR contributes an additional peak centered at 128 ppm resulting from its aromatic carbons. A change in intensity at the C6 region observed between the pEtN enriched cellulose and unmodified cellulose is attributable to the presence of a phosphate moiety in pEtN modified cellulose, Fig. S7a.

As an additional pEtN cellulose detection assay, we analyzed the synthesized product by polysaccharide carbohydrate gel electrophoresis (PACE)[32]. To this end, inverted membrane vesicles (IMVs) were prepared from cells expressing either the wild type *Rs* BcsAB complex alone, or the chimeric or wild type BcsAB complex together with *Ec* BcsG and BcsF. The presence of BcsA and BcsG (when applicable) in the IMVs was confirmed by Western blotting, Fig. 2g. In vitro cellulose biosynthesis reactions were then performed with the IMVs with the expectation that BcsG would modify cellulose using PE lipids as pEtN donors. Following synthesis, the water-insoluble material was isolated after SDS denaturation, and any pEtN units were subjected to modification at the amino nitrogen with N-hydroxysuccinimide (NHS)-conjugated Alexa Fluor 647. This reaction was followed by digestion with cellulase to release water-soluble cello-oligosaccharides, and the released material was analyzed by PACE and imaged (see Methods). Control reactions with either unmodified phosphoric acid swollen cellulose or purified pEtN cellulose from *Ec* only show the release of fluorescently labeled cello-oligosaccharides by cellulase from pEtN cellulose, Fig. S7b, c. This confirms the reliable detection of pEtN cello-oligosaccharides by PACE.

As shown in Figs. 2h and S7d, fluorescently labeled cello-oligosaccharides are readily released by cellulase from the reaction product of the chimeric BcsAB complex in the presence of BcsG. No labeled oligosaccharides are obtained from products produced by wild type *Rs* BcsAB alone. Minor weaker bands are detected when the wild type *Rs* BcsAB complex is co-expressed with BcsG and BcsF. This likely results from modification of cellulose accumulating or precipitating on the membrane surface by the abundantly expressed BcsG. While all three IMV samples produce cellulose in vitro based on ³H-glucose incorporation (Fig. 2i)[6,33], the expression level of the chimeric BcsAB complex is higher, giving rise to approximately two-fold greater cellulose yields, Fig. 2g. To account for limitations in detection levels by PACE, loading approximately twice the amount of the product obtained from the wild type BcsAB complex co-expressed with BcsG and BcsF did not result in stronger PACE signals, Fig S7e. Combined, our NMR and PACE analyses suggest that the close association of BcsG with the BcsA chimera greatly facilitates pEtN cellulose formation.

### The BcsB semicircle tethers a single BcsC subunit

Upon pEtN modification by BcsG, cellulose crosses the periplasm and the OM. This step requires BcsC, a ~130 kDa protein consisting

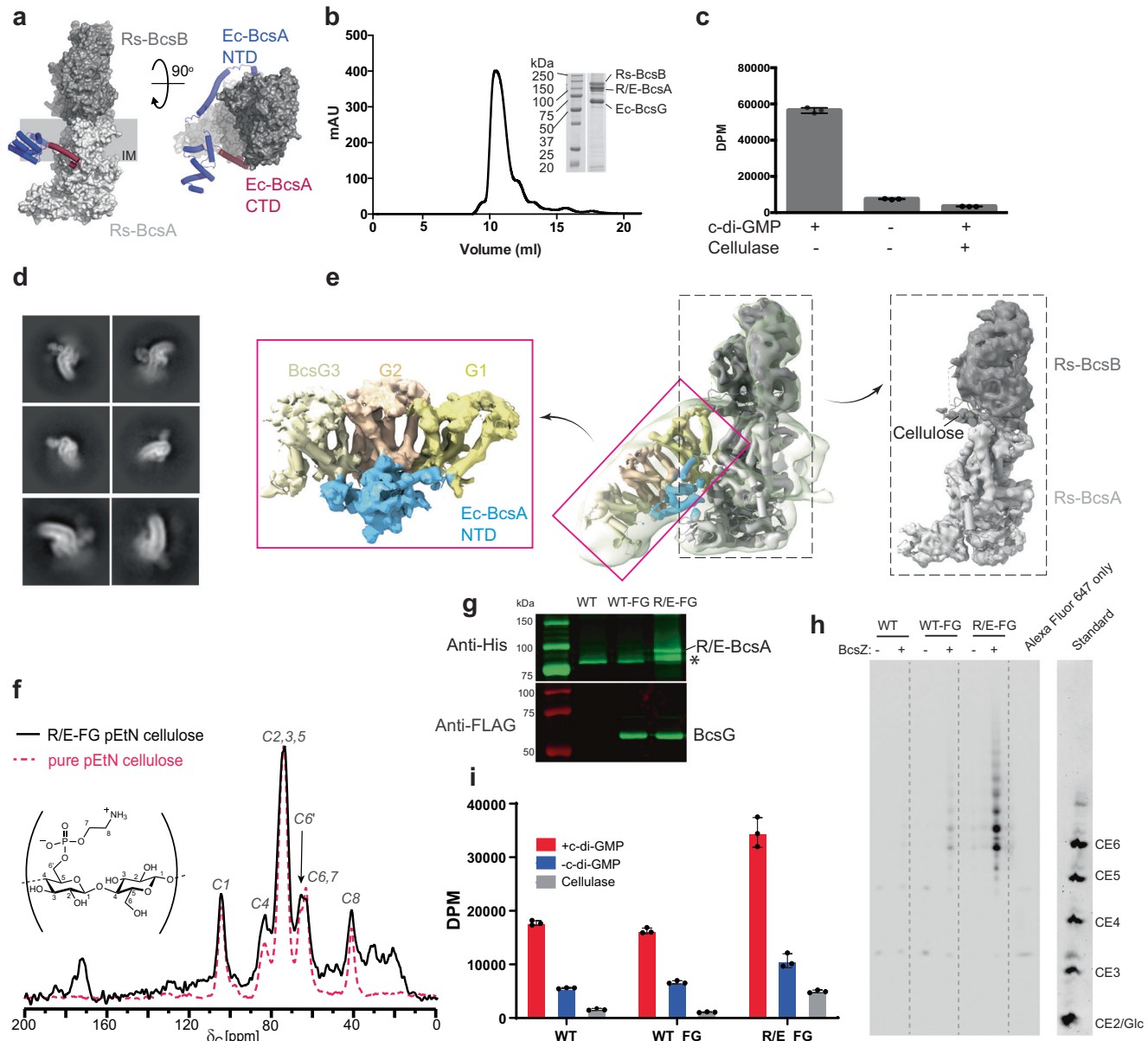

**Fig. 2 | Engineering pEtN cellulose biosynthesis. a** Illustration of the *Rs-Ec* (R/E) BcsA chimera shown as a light and dark gray surface for BcsA and BcsB, respectively (PDB: 4P00). The introduced *Ec* N- and C-terminal domains are shown as cartoons colored blue and red, respectively. **b** Size exclusion chromatography of the chimeric BcsAB-BcsG complex. Inset: Coomassie-stained SDS-PAGE of the peak fraction. **c** In vitro catalytic activity of the purified chimeric BcsAB-BcsG complex. DPM disintegrations per minute. **d** Representative 2D class averages of the chimeric BcsAB-BcsG complex. **e** Low-resolution cryo-EM map (semitransparent surface, contoured at 4.8σ) of the chimeric BcsAB-BcsG complex overlaid with the refined map. Insets show a carved map of the chimeric BcsA NTD-BcsG complex as well as a focused refinement of BcsAB, respectively. *Ec* BcsA NTD is colored blue and trimeric BcsG is colored in shades of yellow. **f** Solid state NMR spectrum of pEtN cellulose produced by the chimeric BcsAB-BcsG complex in vivo (R/E-FG; black line), overlaid with a reference spectrum of pure pEtN cellulose from *Ec* (dashed red line). Peaks labeled C1-8 correspond to the chemical shifts expected for the indicated carbon atoms (see inset). The inset represents the chemical structure of a pEtN modified cellobiose unit with carbon atoms numbered. **g** Western blots of IMVs containing the wild type (WT) *Rs* BcsAB complex alone or the WT or chimeric BcsAB complex (R/E) co-expressed with BcsF and BcsG (FG). BcsA and BcsG are His- and Flag-tagged, respectively. * Indicates an N-terminal degradation product of the chimeric BcsA. **h** Polysaccharide analysis by carbohydrate gel electrophoresis (PACE) of in vitro synthesized pEtN cellulose. IMVs shown in (**g**) were used for in vitro synthesis reactions. Cellulase (BcsZ) released and Alexa Fluor 647-labeled cello-oligosaccharides are resolved by PACE. Cello-oligosaccharide standards are ANTS (8-aminonaphthalene-1,3,6-trisulfonic acid) labeled and range from mono (Glc) to hexasaccharides (CE2-6). This experiment has been repeated twice with similar results. **i** Quantification of cellulose biosynthesis by the IMVs shown in (**g**) containing the indicated BcsA variants and based on incorporation of $^3$H-glucose into the water-insoluble polymer. Error bars in (**c**) and (**i**) represent standard deviations from the means of three replicas. Source data are provided as a Source Data File.

of a C-terminal β-barrel domain preceded by 19 predicted TPR motifs. To investigate the interactions of the IM-associated Bcs complex (IMC) with BcsC by cryo-EM, the purified IMC was incubated with the separately purified N-terminal 18 TPRs of BcsC for

1 h prior to cryo grid preparation (see Methods), Fig. S8a and Table S1.

Cryo-EM analysis of this Bcs complex revealed the previously observed BcsB hexamer architecture associated with one BcsA

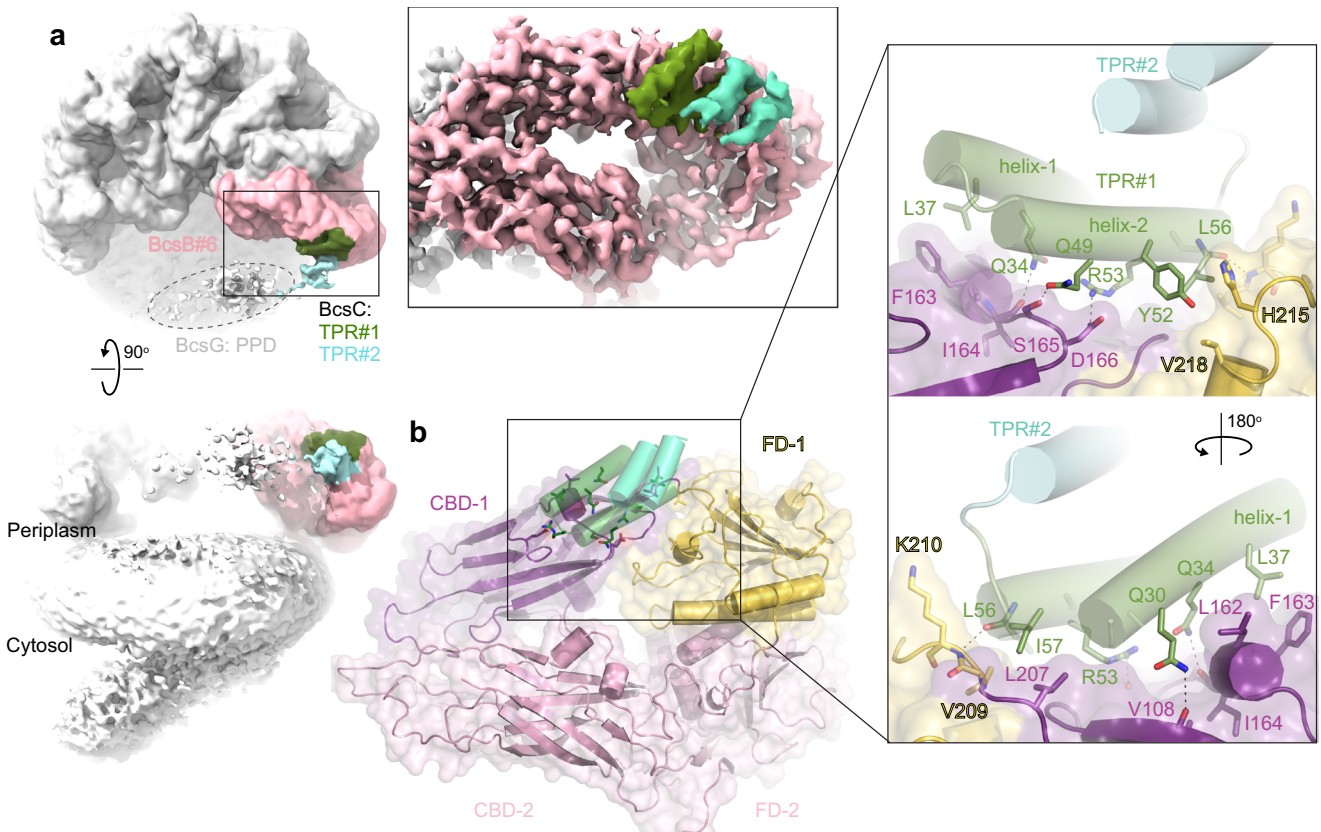

**Fig. 3 | Interactions of BcsC with BcsB. a** Low-resolution map of the Bcs complex in the presence of BcsC's periplasmic domain. The terminal BcsB subunit is colored light pink and BcsC is colored green and light blue for TPR#1 and #2, respectively. TPR tetratricopeptide repeat. The map is contoured at 1.2σ. Inset: High-resolution map of the BcsB-BcsC complex obtained after fusing BcsC's TPRs#1–4 to the N-terminus of BcsB and focused refinement of the BcsB subunit bound to BcsC (contoured at 4.8σ). Density likely representing BcsG's periplasmic domain (PPD) is encircled. **b** Detailed interactions of BcsB and BcsC. FD flavodoxin-like domain and CBD carbohydrate-binding domain.

subunit, Fig. 3a. Viewed from the periplasm and counting clockwise, the first BcsB subunit interacts with BcsA, while the sixth sits at the opposite end of the semicircle (Figs. 1c and 3a). At low contour levels, additional density at the membrane distal tip of the sixth BcsB copy is evident, roughly extending along the perimeter of the semicircle towards the first BcsB protomer, Fig. 3a. The extra density, likely belonging to BcsC's periplasmic domain, is located between BcsB's N-terminal CBD (CBD-1) and the following FD region (FD-1) (referred to the 'TPR binding groove') (Fig. 3a, b). At this contouring, fragmented density at the open side of the semicircle likely representing BcsG's periplasmic domain is also evident near the newly identified BcsC density, Fig. 3a.

Similar to the approach employed for interrogating the BcsA-BcsG interaction, we utilized AlphaFold2 to predict the interactions of BcsB with BcsC's N-terminal TPRs. Using a single copy of BcsB and BcsC's N-terminal TPRs#1–4, AlphaFold2 positions BcsC's TPR#1 with high confidence into BcsB's TPR binding groove, Fig. S8d, e. The predicted model is in excellent agreement with the cryo-EM map. AlphaFold2 did not generate consistent models for a truncated BcsC construct lacking TPR#1, suggesting that the interaction with BcsB indeed depends on the N-terminal region. We conclude that the apical tip of BcsB establishes the interaction with BcsC's TPR#1.

To improve the resolution of the BcsB-BcsC complex map, BcsC's TPR#1–4 were fused to the N-terminus of BcsB, after its N-terminal signal sequence and separated by a linker (see Methods). This fusion construct was co-expressed with all other Bcs components, followed by purification and cryo-EM analyses as described for the wild type Bcs complex. As also observed for the wild type Bcs complex, cryo-EM analysis identified a fully assembled Bcs complex together with (likely

dissociated) partial subcomplexes containing three to five BcsB protomers. Although additional BcsC density similar to the one described above was also observed in a fully assembled complex, the highest quality map revealing the BcsB-BcsC interaction was obtained for a tetrameric BcsB assembly. Nonuniform and focused local refinements generated a map of about 3.2 Å resolution that delineates the specific interactions of BcsB and BcsC, Figs. 3a, b and S3d, S8b, c and Table S1 (see Methods).

The BcsC density associated with BcsB accommodates four α-helices, corresponding to BcsC's N-terminal two TPRs, Fig. 3b. TPR#1 mediates all interactions with BcsB and is best resolved. TPR#2 is rotated by about 45° relative to TPR#1 and extends away from BcsB.

In complex with BcsB, TPR#1 is oriented with its interhelical loop pointing towards the center of the BcsB semicircle. It interacts extensively with one side of CBD-1's jelly roll, as well as the region connecting the jelly roll with FD-1 (residues 208–220), Fig. 3b. This interface contains hydrophobic, polar, and charged residues. In particular, TPR#1's N-terminal helix (helix-1) rests on a short helical segment of a CBD-1 loop, such that its Gln30, Gln34, and Leu37 stack on top of BcsB's Leu162, Phe163, and Ile164. The side chains of Gln30 and Gln34 form hydrogen bonds with the backbone carbonyls of Val108 and Ile164, respectively. The following C-terminal helix of TPR#1 (helix-2) contacts BcsB primarily via polar interactions, including Gln49 and Arg53 that interacts with Ser165 and Asp166 of CBD-1, respectively. The helix's C-terminal end fits into a hydrophobic pocket formed by the CBD-1/FD-1 connection. Here, Leu56 and Ile57 stack against BcsB's Leu207 and Val209, with a backbone hydrogen bond between Leu56 and Lys210 (Fig. 3b).

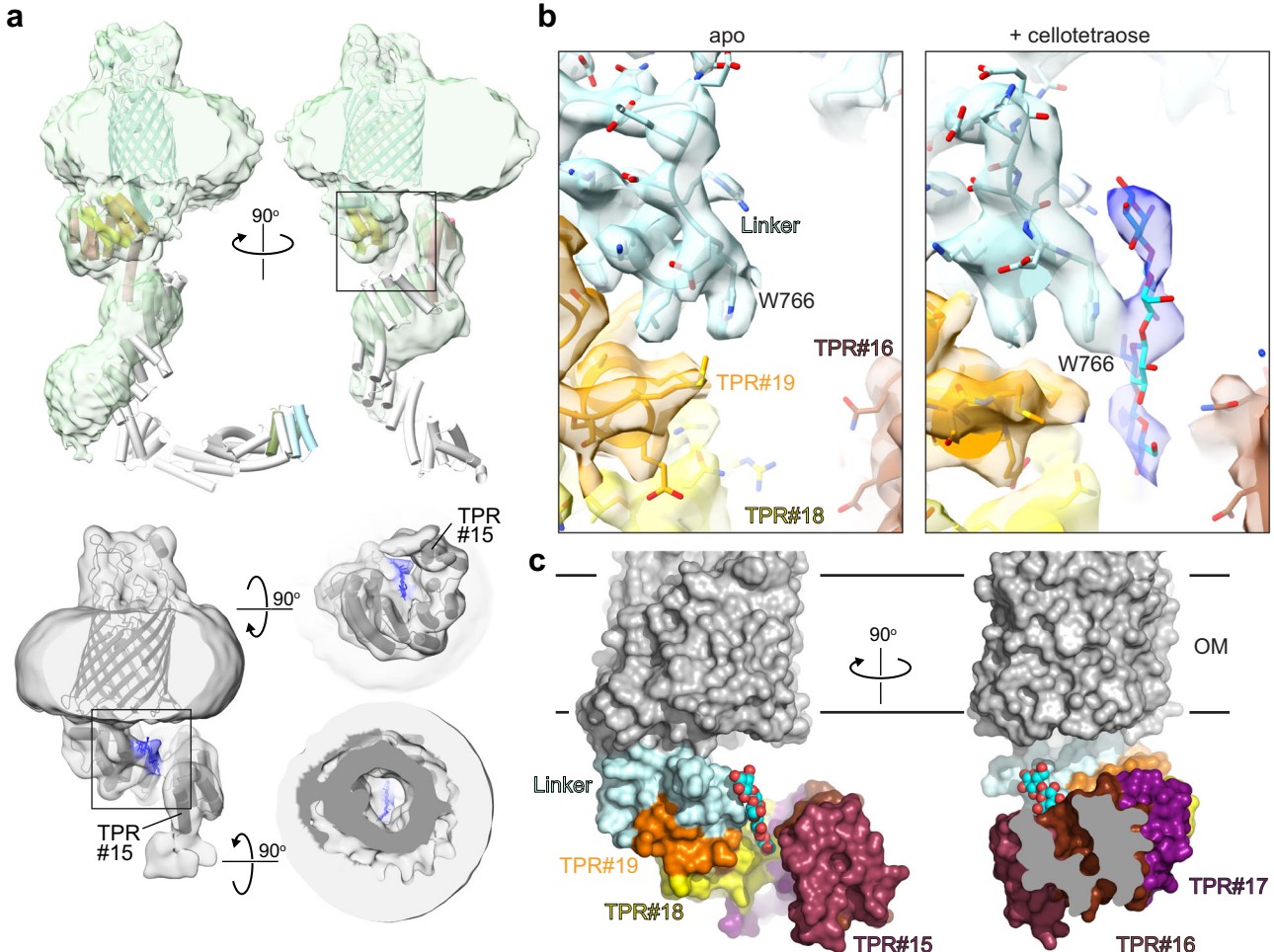

**Fig. 4 | Interactions of BcsC with cellulose. a** Low-resolution cryo-EM maps of BcsC. Top panel: BcsC in the absence of cellotetraose with the full-length Alpha-Fold2-BcsC model (AF-P37650-F1) docked into the density. TPR#1 and #2 are colored green and lightblue, respectively, TPR#15–19 are colored as in (**b**) and (**c**) (TPR tetratricopeptide repeat). Bottom panel: BcsC in the presence of cellotetratose. The model of the refined C-terminal BcsC fragment is docked into the density with the putative ligand colored blue. **b** Close-up views of the putative cellulose binding sites of the refined maps in the absence (apo) and the presence of cellotetraose (shown as sticks colored cyan and red). Both maps are contoured at 10.5σ. **c** Surface representation of BcsC bound to the putative cellotetraose ligand.

The following TPR#2 extends from TPR#1 towards the opening of the BcsB semicircle. Due to this arrangement, the observed BcsB-BcsC interaction is only possible with the last subunit of the BcsB hexamer. Modeling a similar complex with any other subunit of the BcsB semicircle creates substantial clashes between TPR#2 and the FD domains of the neighboring subunit, Fig. S9. This explains why, although present, the fused TPRs of the other BcsB subunits are not resolved.

## BcsC is an outer membrane porin with a periplasmic cellulose-binding solenoid extension

The crystal structure of a C-terminal BcsC fragment containing the β-barrel, a linker, and TPR#19 revealed the porin architecture and its connection with the periplasmic TPR solenoid[8]. Further, crystallographic analysis of the N-terminal six BcsC TPRs from *Enterobacter CJF-002* resolved their solenoid organization[13]. Lastly, the AlphaFold2-predicted model of full-length BcsC supports the solenoid arrangement of its TPR#7–19, forming roughly two helical turns that extend by about 130 Å into the periplasm, Fig. 4a.

To gain experimental insights into the architecture of full-length BcsC and its interaction with cellulose by cryo-EM, we reconstituted the protein into a lipid nanodisc in the absence and the presence of cellotetraose. The obtained cryo-EM maps underscore the high flexibility of BcsC's periplasmic domain. At lower contour levels and for the sample devoid of cellotetraose, the cryo-EM map confirms the

solenoid architecture of TPR#9–19, while the N-terminal eight TPRs are insufficiently resolved or absent in the experimental map, Fig. 4a. Under both conditions, high-resolution refinements (to about 3.2 Å) were only possible for a C-terminal portion of BcsC, beginning with TPR#15 and #16 for the ligand-bound and apo BcsC datasets, respectively, Figs. 4a–c and S3e, f, S10, S11, and Table S1.

The refined BcsC structure is consistent with the AlphaFold2-model, with only minor rigid body translations of TPR#15 towards the solenoid axis, Fig. 4a. Close inspection of the cryo-EM map obtained in the presence of cellotetraose revealed additional elongated density close to the solenoid axis, contacting TPR#16–19, Figs. 4a, b, and S10 and S11. Although the density cannot be identified unequivocally as a cello-oligosaccharide at the current resolution, its shape and interaction with BcsC are consistent with it representing a cellotetraose molecule, perhaps bound in different binding poses. Supporting this interpretation, no additional density at this site or elsewhere is observed in the absence of cellotetraose in a map of similar quality, Figs. 4b and S10. Therefore, we refer to the observed molecule as a 'putative cellulose ligand'.

The most prominent interaction of the putative cellulose ligand with BcsC is mediated by Trp766 at the N-terminus of the linker region, Fig. 4b. The ligand stacks against this aromatic side chain, similar to the cellulose coordination by cellulose synthases and hydrolases[5,34,35]. From here, additional sugar units extend towards the solenoid axis and

contact TPR#18 and #16. Although the putative cello-oligosaccharide is not aligned with the center of the porin channel, Fig. 4a–c, tilting of the glucan chain towards the porin or a different orientation relative to Trp766 could direct it into the OM channel, as described further in the Discussion.

### Cellulase activity is necessary for cellulose secretion

The cellulase BcsZ is a conserved subunit of Gram-negative cellulose biosynthetic systems[3,11]. Similarly, plant and tunicate cellulose synthases are also associated with cellulases for unknown reasons[36,37]. Deleting BcsZ in *Gx* substantially reduces cellulose production in vivo[10,11], while BcsZ has been proposed to reduce biofilm phenotypes on *Salmonella enterica typhimurium*[38].

Accordingly, we sought to delineate whether BcsZ is of similar importance to pEtN cellulose production in *Ec*. To this end, pEtN cellulose secretion from our transformed *Ec* cells was monitored based on CR fluorescence of cells grown on nutrient agar plates, as previously described[20]. This assay takes advantage of substantially enhanced CR fluorescence in the presence of pEtN cellulose, compared to unmodified cellulose[39].

When grown on CR agar plates, *Ec* C43 cells expressing the complete *Ec* Bcs system together with the cyclic-di-GMP producing diguanylate cyclase AdrA, give rise to strong fluorescence, indicative of pEtN cellulose secretion, Fig. 5a. Cells expressing the IMC only, however, reveal background staining, similar to cells producing unmodified cellulose due to the Ser278 to Ala substitution in BcsG[25], Fig. 5a. Consistent with previous observations in *Gx*, CR staining of *Ec* lacking BcsZ indicates substantially reduced pEtN cellulose secretion in the absence of the cellulase, similar to control cells expressing the IMC only, Fig. 5a.

We next investigated whether BcsZ is only required for its cellulose-degrading activity or whether it could be a structural component of the pEtN cellulose secretion system. In the latter case, a catalytically inactive enzyme may still facilitate pEtN cellulose export. To this end, we generated two inactive BcsZ mutants by substituting the catalytic residues Glu55 and Asp243 with Ala[40]. To confirm that the generated BcsZ variants are indeed catalytically inactive, we employed an agar plate-based carboxymethylcellulose digestion assay, as previously described[40]. Here, cellulose digestion by cellulase secreted by plated cells is detected upon CR staining. As expected, *Ec* cells expressing the wild type or the generated BcsZ mutants only show cellulase activity for the wild type enzyme, Fig. S12a, confirming that the generated BcsZ mutants are inactive within the sensitivity limits of the assay. Accordingly, analyzing pEtN cellulose secretion by these cells based on CR fluorescence shows much reduced fluorescence in the presence of the BcsZ: D243A mutant in comparison to the cells expressing *Ec* complex with wild type BcsZ, while no CR staining above background is observed in the presence of the E55A or double BcsZ mutant, Fig. 5a.

Lack of cellulose secretion in the absence of cellulase activity indicates that BcsZ plays a modulating rather than structural role during cellulose export. Accordingly, we probed whether unrelated bacterial cellulases could functionally replace BcsZ. To this end, the *Ec* BcsZ enzyme was replaced in the Bcs expression system with either its *Gx* homolog CMCax (*Gx* produces unmodified fibrillar cellulose)[12], or the Cel9M cellulase domain from the *Clostridium cellulolyticum* cellulosome[41]. Co-expressing the cellulases with the remaining *Ec* Bcs components resulted in detectable cellulase activity in the periplasmic fraction, suggesting that the cellulases were functionally expressed and translocated into the periplasm, Fig. 5b. Monitoring pEtN cellulose secretion by these cells based on CR fluorescence revealed that both cross-species complementations restored secretion by the *Ec* Bcs complex, comparable to wild type levels, Fig. 5c. Further, $^{13}C$ CPMAS NMR was used to evaluate the extent of pEtN modification of cellulose produced in the presence of Cel9M instead of BcsZ. Compared to

products obtained in the presence of BcsZ, the Cel9M sample reveals reduced modification with pEtN, Fig. S12b. Reduced pEtN modification could impact biofilm stability, which will be addressed in the future. Collectively, our results indicate that cellulase activity is critical for cellulose export.

### BcsZ assembles into a tetramer

We failed to detect direct interactions of BcsZ and BcsC biochemically or by cryo-EM analysis. To our surprise, however, we discovered homo-oligomerization of the cellulase. Single particle cryo-EM analysis at a resolution of about 2.7 Å revealed the presence of BcsZ tetramers, in addition to small monomeric particles, Figs. 5d–f and S3g, S12e and Table S1. The tetramers are dimers of homodimers in which the protomers are rotated by about 180° relative to each other, Fig. 5e. The homodimer interface is formed by helices 11 and 12 of the glycosylhydrolase-8 fold. It is rich in ionic interactions, including Asp344 and Arg318 of one protomer and the equivalent residues in the symmetry-related subunit. Similarly, Asp312 interacts with Arg349 across protomers and so does the Asp323 and Arg347 pair. In addition, Gln319 in helix #11 of one protomer hydrogen bonds to the backbone carbonyl oxygen of Gln345 following helix #12 of the symmetry-related subunit.

Two homodimers interact via the N-terminal regions of opposing BcsZ protomers, involving the loop connecting helix 1 and 2 (residues 37–53) as well as a β-strand hairpin connecting helices 3 and 4 (residues 86–114), Fig. 5e, f. This interface also contains several ionic and polar interactions. In particular, Ser104 and Lys105 of the helix3/4 loop are in hydrogen bonding distance to Asn82 in helix 3 of the opposing subunit. Arg41 interacts with Glu39 via a salt bridge, while Gln38 hydrogen bonds to Lys50 across the dimer interface.

Combined, these interactions create a square-shaped tetrameric assembly with the cellulose-binding clefts of BcsZ subunits at opposing corners facing in the same direction. However, because BcsZ binds cellulose in a defined orientation[40], the 2-fold symmetry-related subunits bind their polymeric substrate in opposing directions, Fig. 5e. Of note, the same tetrameric complex was previously observed but not functionally interpreted in apo and cellopentaose-bound BcsZ crystal structures (PDB: 3QXF and 3QXQ respectively), where the described tetramer either represents the crystallographic asymmetric unit or is generated by symmetry mates, Fig. S12c, d[40].

## Discussion

Cellulose is a versatile biomaterial with countless biological and industrial applications. In Gram-negative bacteria, the polysaccharide is secreted across the cell envelope in a single process. Considering cellulose's amphipathic properties, its periplasmic secretion likely requires a shielded translocation path to prevent nonspecific interactions.

Our cryo-EM analysis of the IM-associated Bcs complex reveals a single BcsA cellulose synthase associated with a trimer of the pEtN transferase BcsG. Although present as the full-length enzyme in the analyzed sample, only its membrane-embedded region is resolved in the cryo-EM map. This suggests that the periplasmic catalytic domain is flexibly attached to the TM helices. It is connected to the preceding TM helices via a linker of about 45 residues. In an extended conformation, the linker would allow the TM and periplasmic regions to separate by more than 80 Å, about the height of the BcsB semicircle. However, AlphaFold2 predictions of full-length BcsG reproducibly pack the catalytic domain against the periplasmic corral formed by its interface helices. In this conformation, the catalytic Ser278 points toward the membrane surface where it could receive a pEtN group. The corral may help to position a PE lipid to facilitate this reaction. The in vitro reconstituted pEtN cellulose biosynthesis indeed confirms that *Ec* lipids can serve as the pEtN donor. We hypothesize that following formation of the phosphor-enzyme intermediate with Ser278-bound

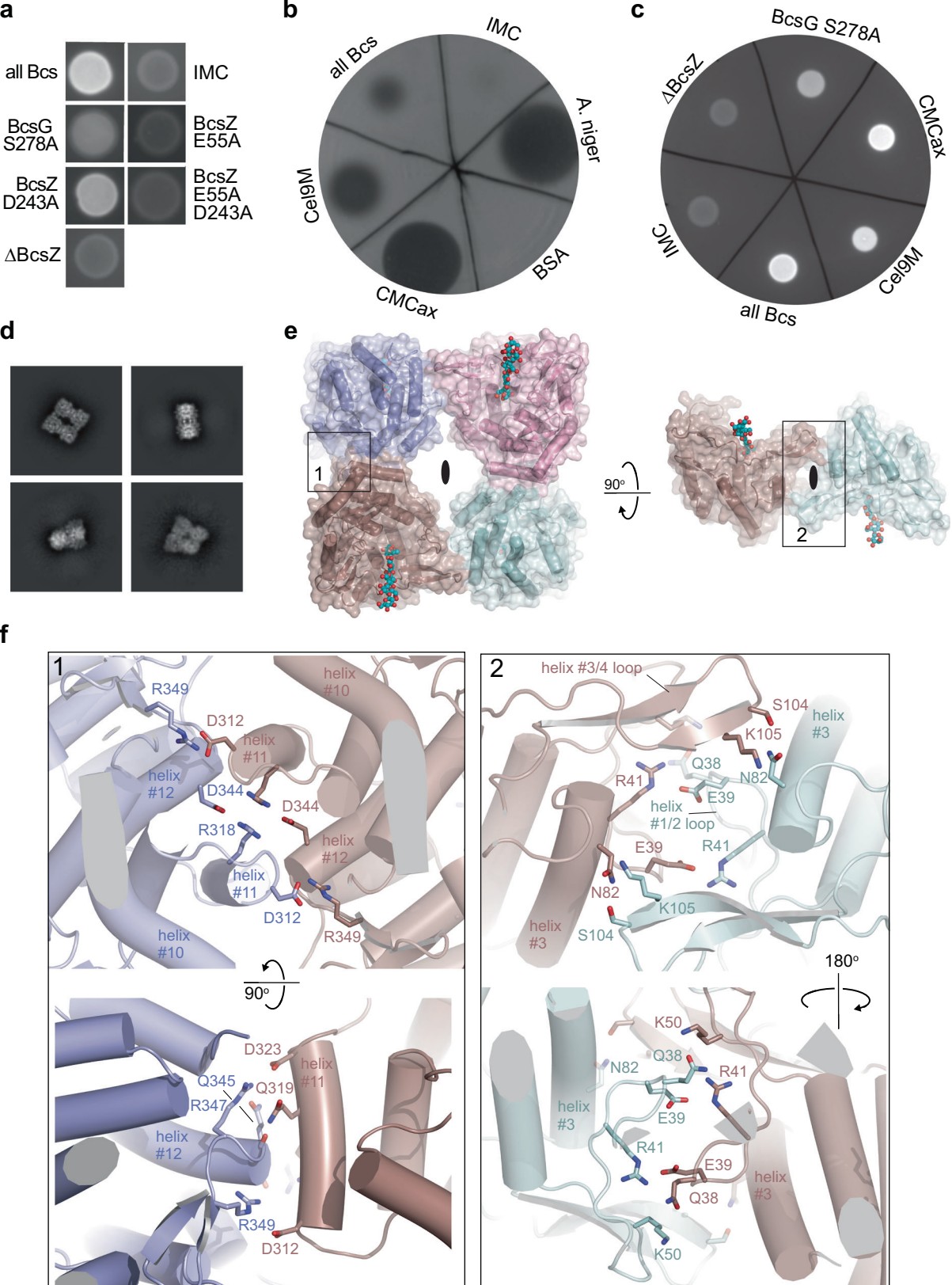

**Fig. 5 | Cellulase activity is necessary for cellulose secretion. a** Congo red (CR) fluorescence image of *Ec* macrocolonies expressing the indicating components as part of the Bcs complex. 'All Bcs' express the inner membrane complex (IMC) together with BcsZ and BcsC. ΔBcsZ: no BcsZ. **b** Carboxymethylcellulose digestion on agar plates using periplasmic *Ec* extracts. Cel9M and CMCax: Periplasmic extracts of cells expressing all Bcs components with BcsZ replaced by the indicated enzyme. BSA and *A. niger*: Controls with purified BSA or *Aspergillus niger* cellulase spotted on the agar plates. Cellulose digestion was imaged after CR staining,

resulting in the observed plaques. **c** Evaluation of CR fluorescence exhibited by *Ec* macrocolonies expressing the indicated components as part of the Bcs complex. **d** Representative 2D class averages of BcsZ tetramers. **e** Model of the BcsZ tetramer shown as a semitransparent surface and cartoon, overlaid with the cellopentaose-bound crystal structure (PDB: 3QXQ, only cellopentaose is shown as ball-and-sticks in cyan and red). Two-fold symmetry axes are indicated by black ellipses. **f** Detailed views of the boxed regions in (**e**). CR and cellulase plate assays were repeated at least three times with similar results. Source data are provided as a Source Data File.

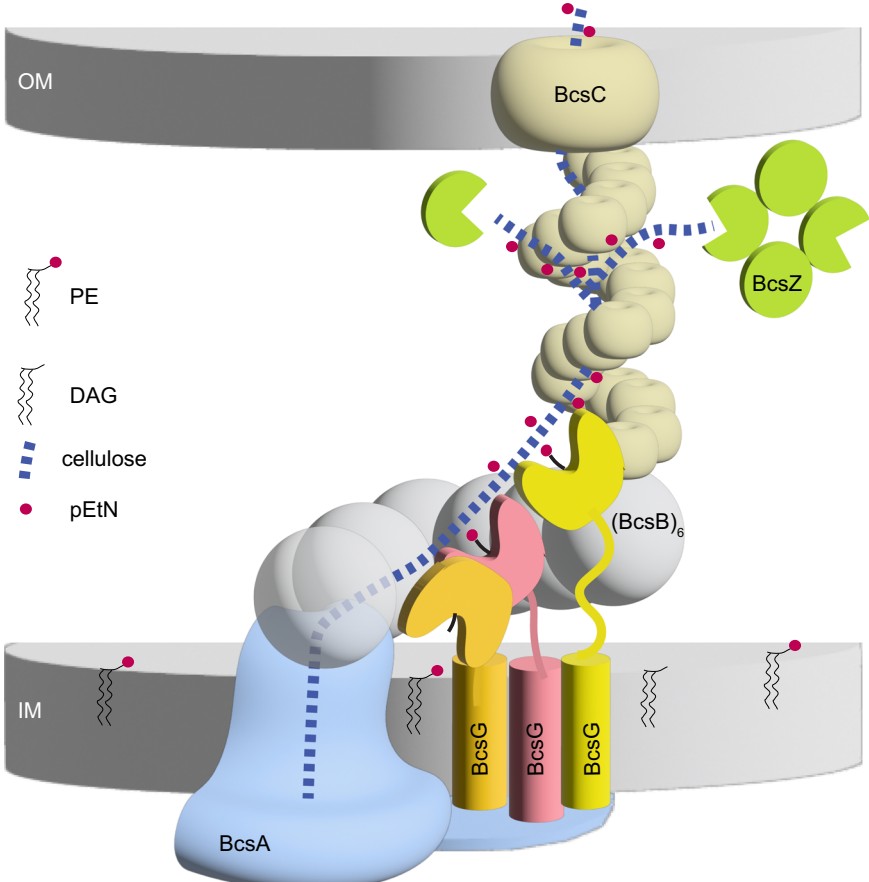

**Fig. 6 | Model of cellulose pEtN modification and secretion.** BcsA recruits three copies of BcsG to the cellulose biosynthesis site via its N-terminal cytosolic domain. The catalytic domain of BcsG either faces the lipid bilayer to receive a pEtN group or contacts the nascent cellulose chain for modification. BcsC interacts with the terminal BcsB subunit of the semicircle to establish an envelope-spanning complex. Cellulose is guided towards the OM through interactions with the TPR solenoid. BcsZ may degrade cellulose to prevent stalling of the biosynthetic machinery or mislocalization of cellulose to the periplasm. The cytosolic BcsE and BcsQR as well as BcsF components are omitted for clarity. PE phosphatidylethanolamine lipid, DAG diacylglycerol, pEtN phosphoethanolamine, IM, and OM inner and outer membrane, respectively.

to pEtN, the catalytic domain disengages from the TM region to access the translocating cellulose polymer. Accordingly, pEtN transfer to cellulose requires an ~90° rotation and substantial translation of BcsG's catalytic domain away from the membrane and towards the nascent polysaccharide.

It is possible that the catalytic domains of the BcsG trimer operate independently, perhaps resulting in stochastic cellulose modification. Previously, it was proposed that two BcsG subunits are necessary to result in pEtN modification of cellulose's C6 hydroxyl groups located on opposing sides of the cellulose polymer[20]. Yet, our cryo-EM maps reveal that three copies are present. A higher copy number could compensate for limiting transfer efficiency.

We also note that the three BcsG periplasmic domains, albeit flexible, essentially close the semicircle formed by the BcsB hexamer. This may facilitate access of BcsG to the cellulose polymer and steer it towards BcsC in the OM. We anticipate that the grafting of *Ec* BcsA's NTD to an unrelated cellulose synthase, allowing it to associate with a BcsG trimer, will unlock the potential to modify cellulose in orthogonal systems for the generation of novel biomaterials.

Because the Bcs complex produces only one cellulose polymer at a time, a single copy of the OM porin BcsC suffices to guide cellulose across the periplasm and the OM. Indeed, the arrangement of BcsC's N-terminal TPRs and steric constraints within a BcsB hexamer ensure that only the terminal BcsB subunit interacts with BcsC. Based on the AlphaFold2-predicted model of full-length BcsC, its 19 TPRs extend by

~150 Å into the periplasm. BcsB's periplasmic domain is about 70 Å tall. Combined, BcsB and BcsC suffice to span the periplasm and the OM, Fig. 6.

The requirement of cellulase activity for efficient cellulose secretion suggests a model in which cellulose hydrolysis facilitates polymer translocation. This could either be by degrading cellulose that is mislocalized to the periplasm, or by overcoming stalling, Fig. 6. Our putative BcsC-cellotetraose complex suggests cellulose translocation along the solenoid, consistent with recent insights into poly N-acetylglucosamine interactions with the TPR of PgaA[42]. Cellulose migrating away from the solenoid helix would likely be irreversibly mislocalized, thereby preventing its proper translocation. In this case, hydrolytic trimming of polymers accessible from the periplasm could reset the translocation process, Fig. 6. The resulting cello-oligosaccharides may either remain in the periplasm or be imported for degradation to avoid osmotic destabilization, as observed in *Pseudomonas aeruginosa* upon accumulation of periplasmic alginate[43]. Similarly, stalling of cellulose secretion due to interactions with other surface polymers could be alleviated by polymer cleavage in the periplasm, followed by diffusion into the extracellular matrix. Oligomerization of BcsZ may optimize the enzyme's ability to interact with and digest cellulose to increase catalytic efficiency, as frequently observed for carbohydrate-active enzymes[44].

Taken together, our analyses suggest the synthesis and translocation of a single cellulose polymer by the *Ec* Bcs complex. The positioning of the first BcsB subunit 'above' the cellulose secretion channel

steers the nascent chain towards the center of the BcsB semicircle, past the BcsG trimer, Figs. 6 and S13. Flexibility of BcsG's catalytic domain enables contacts with membrane lipids to recognize and attack a pEtN group as well as accessing the translocating cellulose polymer at different distances from BcsA. The modified cellulose polymer likely migrates along the TPR solenoid until reaching the OM channel. Polymers 'escaping' into the periplasm or jamming the secretion system would be cleaved by BcsZ, Fig. 6. This model explains why BcsZ's hydrolytic activity is necessary to facilitate cellulose secretion and why the enzyme can be replaced with off-the-shelf cellulases. Hydrolytic clearance of roadblocks may also assist cellulose microfibril formation in other kingdoms of life.

## Methods

### Construct design and mutagenesis

**BcsA NTD and BcsG.** The N-terminal region of BcsA corresponding to the first 94 residues (NTD) was amplified from the existing pETDuet_Ec_Bcs_A-12His_nSS-Strep-B_AdrA-6His plasmid (pETDuet_Ec_Bcs_AB_AdrA)[20] with a primer-encoded C-terminal Strep-tag II (Table S2) and cloned into a pETDuet-1 vector using NcoI and XhoI restriction sites. BcsB has its native signal sequence (nSS) in this plasmid. The *BcsG* gene was amplified from an existing pACYCDuet_Ec_Bcs_PelB-8His-C-FLAG_PelB-Z_F_G-FLAG plasmid (pACYCDuet_Ec_Bcs_CZFG)[20] with a C-terminal dodecahistidine tag (primers listed in Table S2) and cloned into the vector pACYCDuet-1 using NcoI and XhoI restriction sites.

**BcsA chimera.** The *Rs-Ec* (R/E) BcsA chimera was engineered by fusing the N-terminus of *Ec* BcsA (1–149) with *Rs* BcsA (17–728) and replacing the *Rs*-BcsA C-terminus (729–788) with the *Ec* BcsA extended C-terminus (828–872). The BcsA chimera was synthesized by Gene Universal and cloned into the pETDuet-1 vector using NcoI and HindIII restriction sites, which inserted an additional Gly residues after the N-terminal Met. Next, *Rs* BcsB was inserted into the second cloning site of the pETDuet-1 vector using NdeI and KpnI restriction sites, generating plasmid pETDuet_Rs_Ec_Bcs_A-12His_Rs_Bcs_B.

**AdrA.** For the purification of in vivo synthesized cellulose, c-di-GMP generating enzyme, AdrA was amplified from the pETDuet_Ec_Bcs_AB_AdrA plasmid[20] using primers listed in Table S2 and inserted into the empty pCDFDuet-1 vector using restriction enzyme digestion and ligation cloning (pCDFDuet_AdrA).

**BcsB-BcsC fusion.** For generating the BcsB-BcsC fusion construct, the pETDuet_Ec_Bcs_AB_AdrA plasmid was used. The N-terminal four TPRs of BcsC (TPR#1–4; residues 21–179) followed by a GSGSGS linker were inserted by polymerase incomplete primer extension (PIPE) cloning after the N-terminal signal sequence and Strep-tag II of BcsB, followed by residues 55-779 of BcsB. This generated plasmid pETDuet_Ec_Bcs_A-12His_nSS-Strep-TPR(#1-4)-(GS)$_3$-B_AdrA.

**BcsC.** BcsC's periplasmic domain (TPR#1–18; residues 24–709) was amplified from the existing full-length BcsC plasmid[8] using primers listed in Table S2 and cloned into a pET20b vector between NcoI and XhoI restriction sites. The full-length construct contained an N-terminal and the periplasmic domain a C-terminal deca His-tag respectively.

**BcsZ.** Mutagenesis of BcsZ was done in pACYCDuet_Ec_Bcs_CZFG plasmid using the QuikChange approach (primers listed in Table S2). Deletion of BcsZ was also done using the same pACYCDuet_Ec_Bcs_CZFG plasmid by PIPE cloning. To replace *Ec* BcsZ with other cellulases, the *Gx* BcsZ (CMCax) and *C. cellulolyticum* Cel9M genes were synthesized with the DsbA and wild type *Ec* BcsZ signal sequences, respectively. The genes were inserted into the previously described pACYCDuet_Ec_Bcs_CZFG vector using PIPE cloning. This resulted

in the generation of pACYCDuet_Ec_Bcs_C_SS-Cel9M/CMCax_FG plasmid respectively.

**BcsG.** The catalytically inactive S278A mutant of BcsG was also generated by QuikChange mutagenesis using the pACYCDuet_Ec_Bcs_CZFG plasmid[20].

### Protein expression and purification

**BcsA NTD and BcsG expression and tandem purification.** The Strep-tagged BcsA NTD and His-tagged BcsG were co-expressed in *Ec* C43 (DE3) cells in Terrific Broth-M-80155 media (TB-AD)[20] containing 100 µg/mL ampicillin and 35 µg/mL chloramphenicol. 11 L of bacterial cell cultures were grown at 37 °C until the cell density reached OD$_{600}$ of 0.8 at which point the temperature was lowered to 20 °C and growth was continued for another 18 h. Cells were harvested by centrifugation for 20 min at 5000 rpm and 4 °C. Pelleted cells were resuspended in ice-cold Buffer A (25 mM Tris pH 8.0, 300 mM NaCl, 5% glycerol) containing 1 mM PMSF and 1x protein inhibitor cocktail (0.8 µM Aprotinin, 5 µM E64, 10 µM Leupeptin, 15 µM Bestatin-HCl, 100 µM AEBSF-HCl, 2 mM Benzamidine-HCl and 2.9 mM Pepstatin A). The cells were lysed using a gas powered microfluidizer (25 kpsi, 3 passes) and the lysates were spun at $20,000 \times g$ for 20 min at 4 °C in a JA-20 rotor (Beckman) to remove cell debris. The membrane containing supernatant was collected and subjected to ultracentrifugation in a Ti45 rotor (Beckman) at $200,000 \times g$ for 2 h at 4 °C. Membrane pellets were flash frozen in liquid nitrogen and stored in -80 °C until used.

To test the interaction between BcsA NTD and BcsG, a tandem affinity purification (TAP) scheme was followed. To this end, membranes were first solubilized in Buffer A containing Detergents A [1% lauryl maltose neopentyl glycol (LMNG, Anatrace), 0.2% decyl maltose neopentyl glycol (DMNG, Anatrace) and 0.2% cholesteryl hemisuccinate (CHS, Anatrace)], 40 mM imidazole, 1 mM PMSF and 1x protein inhibitor cocktail. After incubation for 1 h at 4 °C with mild agitation, non-solubilized material was removed by centrifugation at $200,000 \times g$ for 40 min at 4 °C. During this time, 7 mL of His-Pur Ni-NTA resin (Thermo Scientific) was equilibrated in Buffer A containing 40 mM imidazole. The membrane extract was applied to these pre-equilibrated beads and allowed to gently rock for 1 h at 4 °C. The resin was transferred to a gravity flow column and washed two times with Buffer A containing Detergents B (0.01% LMNG, 0.002% DMNG, 0.002% CHS) supplemented with 40 and 50 mM imidazole respectively. The protein was eluted with 400 mM imidazole and the eluent was immediately diluted with Buffer A containing Detergents B to dilute the imidazole to 200 mM. During this Ni-NTA chromatography, Strep-Tactin resin (IBA) was also equilibrated with Buffer A containing Detergents B. The diluted eluent was passed over the Strep-Tactin resin twice at room temperature, followed by 5 column volumes of wash with Buffer A containing Detergents B and eluted with Buffer A containing 3 mM desthiobiotin and Detergents B. The eluent was concentrated and loaded onto Superose 6 increase 10/300 GL column (GE Healthcare) equilibrated with Buffer B (25 mM Tris pH 8.0, 100 mM NaCl) containing Detergents C (0.003% LMNG, 0.0006% DMNG, 0.0006% CHS). Initially the gel filtration fractions were run on SDS-PAGE for Coomassie staining and finally, the interactions between these two components were confirmed by Western blotting and tandem mass spectrometry fingerprinting.

**Chimeric BcsAB-BcsG complex.** To co-express the BcsA-BcsB chimera complex with BcsG and BcsF, freshly transformed *Ec* C43 (DE3) cells containing the pETDuet_Rs_Ec_Bcs_A-12His_Rs_Bcs_B along with the pACYCDuet_Ec_Bcs_FG-FLAG plasmid (pACYCDuet_Ec_Bcs_FG)[20] were grown in four times 1 L of TB-AD media using the above described protocol. After harvesting the cells, pellets were resuspended to a final volume of 250 mL in Buffer C (25 mM HEPES pH 8.0, 300 mM NaCl, 5 mM cellobiose, 5% glycerol, and 5 mM MgCl$_2$) using a glass dounce

homogenizer and lysed using a microfluidizer (25 kpsi, 3 passes) in the presence of 1 mM PMSF and protein inhibitor cocktail (as described above). After pre-clearing the lysates by a low-speed centrifugation step, membranes were collected and stored as described above.

Membranes were resuspended in solubilization buffer containing Buffer C, mix of Detergents (Detergents A), protease inhibitor cocktail, and 1 mM PMSF for 1 h at 4 °C on a rotating shaker. Insoluble material was removed by ultracentrifugation at 200,000 × g for 40 min and the supernatant was incubated with gentle rocking for 1 h at 4 °C with 5 mL of His-Pur Ni-NTA resin equilibrated in buffer C containing 40 mM imidazole. After batch binding, the resin was packed into a gravity flow column and washed with Buffer C containing Detergents B and 40 mM imidazole. Following this, three more washes were done with Buffer C containing Detergents B supplemented with 50 mM imidazole, 700 mM NaCl, or 55 mM imidazole, respectively. Protein was eluted with Buffer C containing Detergents B and 400 mM imidazole. The protein eluent was concentrated to 500 μL and subjected to size-exclusion chromatography on a Superdex 200 increase 10/300 GL column (GE Healthcare) equilibrated with Buffer D (25 mM HEPES pH 8.0, 150 mM NaCl, 5 mM MgCl$_2$, 0.5 mM cellobiose) containing Detergents C. The sample quality was evaluated by peak shape, SDS-PAGE and negative stain EM. The same protocol was used to co-express and purify BcsG with the wild type $Rs$ BcsAB (pETDuet_Rs_Bcs_A-12His_B[6]) complex.

**Congo red binding and fluorescence assays.** For CR assays, the $Ec$ Bcs total membrane complex (TMC; including inner and outer membrane components) constituting the functional cellulose synthase machinery was expressed using all three plasmids (pETDuet_Ec_Bcs_AB_AdrA, pACYCDuet_Ec_Bcs_CZFG, and pCDFDuet_Ec_Bcs_R_Q_HA-E) and the $Ec$ Bcs_inner membrane complex (IMC) was expressed using the pACYCDuet_Ec_Bcs_FG plasmid instead of pACYCDuet_Ec_Bcs_CZFG, along with the other two plasmids (pETDuet_Ec_Bcs_AB_AdrA and pCDFDuet_Ec_Bcs_R_Q_HA-E). Different versions of BcsZ or the BcsG mutant in the pACYCDuet_Ec_Bcs_CZFG plasmid generated in this study (as described above in construct design) were also expressed like the TMC complex using the two other pET and pCDFDuet_Ec_Bcs plasmids.

**Bcs complex purification.** The $Ec$ Bcs complex was expressed and purified as described previously[20]. Briefly, the three plasmids used to express TMC complex were transformed into $Ec$ C43 (DE3) cells followed by expression in 4 × 1 L of TB-AD media containing the necessary antibiotics. After the membrane preparation, the complex was purified using Ni-NTA affinity chromatography followed by Strep-Tactin resin purification. The strep eluent was loaded onto a Superose 6 increase 10/300 GL column and the inner membrane complex (IMC) eluted in a sharp peak roughly at 13 mL elution volume. The fractions containing the $Ec$ Bcs complex were pooled and used for reconstitution into nanodiscs at a 1:4:160 molar ratio of IMC:MSP2N2:$Ec$ total lipid extract (solubilized in 100 mM sodium cholate). Gel filtration buffer (50 mM HEPES pH 8.0, 150 mM NaCl, 5 mM MgCl$_2$, 0.5 mM cellobiose containing Detergents C), lipid, sodium cholate (15 mM final concentration) and detergent-solubilized IMC was combined and incubated for 1 h at 4 °C with gentle rocking to form the mixed micelles. MSP2N2 was added and incubated for another 30 min at 4 °C. BioBeads (Bio-Rad) were added stepwise (three times) in equal mass to remove detergent; first, after the MSP2N2 addition, followed by a second addition after 1 h and the third addition 12 h later. Nanodisc-reconstituted IMC complex was purified on a Superose 6 increase 10/300 GL column equilibrated in gel equilibration buffer with no detergents. The reconstituted fractions were screened using SDS-PAGE and negative-stain EM for the presence of MSP2N2 and IMC.

For the $Ec$ complex with the engineered BcsB-BcsC fusion, $Ec$ C43 cells were co-transformed by electroporation with pETDuet_Ec_Bcs_A-12His_nSS-Strep-TPR(#1-4)-(GS)$_3$-B_AdrA along with two other plasmids,

namely pACYCDuet_Ec_Bcs_CZFG and pCDFDuet_Ec_Bcs_R_Q_HA-E. The complex was expressed and purified as described for the wild type $Ec$ Bcs complex without the nanodisc formation.

**BcsC TPR purification.** To purify the periplasmic domain of BcsC, freshly transformed $Ec$ Rosetta 2 cells were grown in autoinducing TB-AD media for 25 h at 28 °C. The cells were harvested by centrifugation at 5000 rpm for 20 min. The periplasmic extract was prepared in Tris/EDTA/Sucrose (TES) buffer as described[45]. The protein was purified from the periplasmic extract via immobilized metal affinity chromatography on Ni-NTA agarose resin. The isolated extract was dialyzed overnight against buffer E (25 mM Tris pH 7.5 and 100 mM NaCl) and loaded onto Ni-NTA beads equilibrated with the buffer E. After batch binding for 1 h, the resin was washed with Buffer E and Buffer E supplemented with 30 mM imidazole, 500 mM NaCl or 40 mM imidazole. The protein was eluted with 400 mM imidazole and concentrated and applied to Superdex 200 increase 10/300 GL column (GE Healthcare) equilibrated in 0.2 M sodium bicarbonate buffer (pH 7.5), 500 mM NaCl. The peak fraction containing the protein was concentrated and aliquots were flash frozen for future use.

**BcsC purification.** For full-length BcsC, membranes were prepared and purification was carried out as described previously for the $Ec$ BcsC2 porin construct[8] with some modifications. Briefly, BcsC membranes were solubilized in Buffer F containing 25 mM Tris pH 8.5, 300 mM NaCl, 5% glycerol, 35 mM imidazole, 30 mM LDAO (lauryldimethylamine-N-oxide, Anatrace) and 3 mM DDM (dodecyl-β-D-maltopyranoside, Anatrace) for 1 h. After removal of the insoluble aggregates by centrifugation at 200,000 × g for 30 min, the supernatant was combined with 5 mL Ni-NTA beads equilibrated in Buffer F containing no detergents and allowed to rock for 1 h at 4 °C. Resin was collected in a gravity flow column and washed with buffer F without LDAO but substituted with 1 mM DDM and 40 mM imidazole, 1 M NaCl, or 50 mM imidazole. Protein was eluted using buffer G containing 25 mM Tris pH 8.5, 100 mM NaCl, 300 mM imidazole, and 0.6% C8E4 (tetraethylene glycol monooctyl ether, Anatrace) and concentrated before loading onto a Superdex 200 increase 10/300 GL column (GE Healthcare) equilibrated with buffer G containing no imidazole. The fractions containing BcsC were concentrated and reconstituted into MSP1E3D1 nanodiscs with $Ec$ total lipids (solubilized in sodium cholate) at a final molar ratio of 1:4:100, as described above for the $Ec$ Bcs complex, in the absence or presence of 5 mM cellotetraose (Megazyme). After detergent removal, the sample was purified over the Superdex 200 increase size exclusion column equilibrated in 25 mM Tris pH 8.5, 100 mM NaCl. Peak fractions corresponding to BcsC in MSP1E3D1 were collected and the sample quality was evaluated by SDS-PAGE and negative stain EM.

**BcsZ purification.** BcsZ was expressed and purified as described previously using an existing plasmid (pET20b_PelB_BcsZ_6His)[40]. The protein was purified using Ni-NTA and size-exclusion chromatography.

**Cellulose synthase enzyme assays**
Cellulose biosynthesis assays were performed as described previously[6]. This assay measures the incorporation of UDP-[³H]-glucose into insoluble glucan chains. Briefly, 20 μL reaction was performed in the respective gel filtration buffer by incubating the enzyme in the presence of 20 mM MgCl$_2$, 5 mM UDP-glucose (UDP-Glc, Sigma, Cat# U4625), 0.25 μCi UDP-[³H]-Glc (PerkinElmer, Cat# NET1163250UC), 30 μM c-di-GMP at 30 °C for the chimeric BcsAB-BcsG complex for 1 h at an enzyme concentration of 1 mg/mL. For IMVs, assays were carried out in a similar manner but incubated for 16 h at 37 °C. Following biosynthesis, the reaction mixture was spotted onto the origin of a descending Whatman-2MM chromatography paper, which was developed with 60% ethanol. The high molecular weight

polymer retained at the origin was quantified on a liquid scintillation counter (Beckman). Control reactions were set up by replacing the c-di-GMP with ddH$_2$O. To confirm the formation of authentic cellulose, another set of reaction was set up wherein 5 U of endo-(1,4)-β-gluca-nase (E-CELTR; Megazyme) was added at the beginning of the synthesis reaction for the enzymatic degradations of the in vitro synthesized glucan. Each condition was performed in triplicate and error bars represent deviations from the means.

## Cellulase assay

In order to examine the cellulase activity of the BcsZ mutants as well as the new cellulases (Cel9M and CMCax), a cellulase activity assay was performed using Carboxymethyl cellulose agar plates, as described previously[40]. CMC-agar plates were prepared by dissolving 2% CMC and 1.5% agar in LB medium, followed by autoclaving. The solution was cooled and supplemented with 0.5 mM isopropyl β-D-thiogalactopyranoside (IPTG) and respective antibiotics prior to pouring the plates. To test for cellulase activity of Cel9M and CMCax, these proteins were co-expressed along with the *Ec* Bcs TMC complex in C43 cells using the pACYCDuet_Ec_Bcs_C_SS-Cel9M/CMCax_FG plasmid instead of pACYCDuet_Ec_Bcs_CZFG plasmid and periplasmic fractions were extracted as described above for the purification of BcsC's periplasmic domain. Negative and positive controls were performed by spotting commercially available purified bovine serum albumin (BSA) or *Aspergillus niger* cellulase (Sigma) onto the plates. CMC-agar plates were incubated at 37 °C for 48 h and stained with 2% CR solution for 1 h at room temperature, followed by destaining in 1 M NaCl for 2 h.

To probe for BcsZ cellulase activity in the BcsZ mutants, the pACYCDuet_Ec_Bcs_C_Z$_{wt/mutants}$FG plasmid was transformed in C43 cells and plated on LB-agar plates containing 25 μg/mL chloramphenicol. After an overnight incubation of the plates at 37 °C, a single colony was picked from each plate to inoculate 5 mL LB broth and grown overnight at 37 °C. Next day, all cultures were normalized based on OD$_{600}$ absorbance, and 5 μL from each culture was spotted onto the CMC-agar plates. After incubating the agar plates at 37 °C for 48 h, colonies were removed from the plates prior to staining with CR as described above. All cellulase plate assays were performed in triplicates.

## Congo red binding and fluorescence assays

Starter cultures of *Ec* complex transformed cells were grown overnight at 37 °C in a shaking incubator in LB medium. The overnight cultures were normalized to an optical density at 600 nm (OD$_{600}$) of 1 with sterile fresh LB medium and 5 μL of this diluted culture was spotted onto the LB agar plates lacking NaCl but containing 25 μg/mL CR, 250 μM IPTG and the antibiotics ampicillin, chloramphenicol and streptomycin. The agar plates were kept at room temperature for 48–56 h and the bacterial cells on top of the agar were visualized using G:Box Chemi-XX6 (Syngene, Cambridge, UK). Images were acquired with GeneSys software (Syngene, version 1.8.5.0) based on the excitation and emission wavelength of the CR (497/614 nm). All experiments were performed in triplicate.

## Purification of in vivo synthesized pEtN cellulose

For the purification of in vivo synthesized cellulose, *Ec* C43 cells were co-transformed with pETDuet_Rs_Ec_Bcs_A-12His_Rs_Bcs_B together with the pAYCDuet_Ec_Bcs_FG and pCDFDuet_AdrA plasmids. Peri-plasmic cellulose produced by the BcsA chimera was obtained through cell lysis; digestion of DNA, RNA, and protein; and precipitation and purification of polysaccharide. First, four 1-L cell cultures were grown in TB-AD media with slow shaking for 25 h at 28 °C. Cells were collected by centrifugation at 5000 *g* for 20 min, flash frozen in liquid nitrogen, and stored at -80 °C until processed. To isolate pEtN cellulose, cells were thawed, resuspended in lysis buffer (10 mM Tris pH 7.4, 0.1 M

NaCl, and 0.5% SDS), sonicated briefly, and treated with lysozyme (final concentration of 5 mg/mL) with rocking at room temperature (RT) for 30 min. This suspension was then subjected to boiling with constant stirring for 1 h and then cooled to room temperature. The suspension was then treated with DNase and RNase (each at final concentrations of 100 μg/mL) and incubated at RT for 1 h. Trypsin and chymotrypsin were then added (each at final concentration of 50 μg/mL), followed by rocking at RT for 4 h. Lastly, Proteinase-K (final conc 100 μg/mL) was added and this suspension was incubated overnight with moderate shaking at 60 °C. The solution was then cooled to RT and diluted with Milli-Q (MQ) water to reduce the final SDS concentration to less than 0.2%. This solution was subjected to dialysis against MQ water for 24 h using 100 kDa cut-off cellulose ester dialysis membranes. Then the solution was subjected to one freeze-thaw cycle. After thawing, CR (25 μg/mL) and NaCl (170 mM) were added while stirring to facilitate purification and precipitation of pEtN cellulose from the largely clarified lysate. The solution was transferred to 50 mL falcon tubes and insoluble material was pelleted via centrifugation at $13,000 \times g$ for 1 h to collect the enriched pEtN cellulose. The pellet was further washed with 4% SDS and 10 mM Tris pH 7.4 followed by brief sonication. The solution was allowed to rock at RT overnight, followed by centrifugation for 2 min at $13,000 \times g$ to pellet the cellulosic material. The final pellet was washed with MQ water 3–5 times to remove the SDS with pelleting by centrifugation following each resuspension. The final sample was frozen and lyophilized and used for NMR analysis. Unmodifed cellulose as a control was purified using CR as a purification aid[1].

pEtN cellulose was also isolated from C43 cells expressing the *Ec* complex with either wild type BcsZ or BcsZ being replaced with Cel9M, as described previously[20]. Briefly, four 1 L cell cultures expressing the *Ec* Bcs complex were grown in TB-AD media with slow shaking for 24 h at 28 °C. The cells were harvested by centrifugation, resuspended in 10 mM Tris pH 7.4, and sheared using a homogenizer on ice. Following shearing, the cells were removed by three rounds of centrifugation at $10,000 \times g$ for 10 min and the supernatant was dialyzed against MQ water for 24 h. The dialyzed solution was then frozen and thawed, and the insoluble material was pelleted by centrifugation at $13,000 \times g$. The obtained pellet was treated with 4% SDS in Tris buffer overnight and subsequently washed to remove the SDS. The resulting cellulose was lyophilized and analyzed via solid-state NMR.

## Solid-state NMR analysis

$^{13}$C CPMAS solid-state NMR was performed at ambient temperature in an 89 mm bore 11.7 T magnet (Agilent Technologies, Danbury, CT) using an HCN Agilent probe with a DD2 console (Agilent Technologies)[46]. Samples were spun at 7143 Hz in 36 μL capacity 3.2 mm zirconia rotors. CP was performed with a field strength of 50 kHz for 13 C and with a 10% linearly ramped field strength centered at 57 kHz for $^1$H. $^1$H decoupling was performed with two pulse phase modulation (TPPM) at 83 kHz[47]. The experimental recycle time was 2 s. $^{13}$C chemical shift referencing was performed by setting the high-frequency adamantane peak to 38.5 ppm[48]. The enriched pEtN cellulose sample obtained from the BcsA (*Rs/Ec*) chimera co-expressed with BcsG and BcsF was 3 mg and the $^{13}$C CPMAS spectrum is the result of 40,960 scans.

Similarly, the sample size for the wild type *Ec* complex with BcsZ was 7.1 mg, while for Bcs complex with Cel9M, the sample size was 3.2 mg and each $^{13}$C CPMAS spectrum is the result of 40,960 scans. The spectra are scaled by the anomeric peak intensity to enable facile comparison of the extent of pEtN modification between samples.

## Preparation of inverted membrane vesicles

IMV preparation was carried out as described previously[49] either for the wild type *Rs* BcsAB complex alone, or the chimeric BcsAB or wild type BcsAB complex co-expressed with *Ec* BcsG and BcsF. Briefly, the complex along with AdrA was overexpressed in *Ec* C43 cells in TB

media. When the cell density reached $OD_{600}$ of 0.8, expression was induced by addition of 0.6 mM IPTG. After 4 h of incubation at 37 °C, cells were harvested and resuspended in buffer H containing 20 mM phosphate buffer (pH 7.5) and 100 mM NaCl. The cells were lysed in a microfluidizer and cell debris was removed by low-speed centrifugation (20,000 × $g$; JA-20 rotor). The supernatant (~22 ml) was layered over a 2 M sucrose cushion and centrifuged in a Ti-45 rotor at 200,000 × $g$ for 2 h. Following this, the dark brown ring formed at the sucrose interface was carefully collected (~10 ml) and diluted to 65 mL in Buffer H and the membrane vesicles were sedimented by centrifugation at 200,000 × $g$ for 90 min. The pellet was then rinsed and resuspended in 1 mL of Buffer H, homogenized using a 2 mL Dounce homogenizer, and stored in aliquots at -80 °C. The expression of BcsA and BcsG was detected by Western blotting using Anti-His and Flag antibodies, respectively.

## Analysis of phosphoethanolamine modification by PACE

IMVs were used for synthesizing cellulose in vitro[49]. 500 µL of reactions were set up by incubating IMVs in the presence of 20 mM $MgCl_2$, 5 mM UDP-Glc, and 30 µM c-di-GMP at 37 °C for 16 h. The amount of IMVs used was standardized based on radiometric cellulose quantification (as described above). Following the incubation, the reaction was terminated with 2% SDS, and the insoluble polymer was pelleted by centrifugation at 21,200 × $g$ at room temperature. The obtained pellet was washed four times with water to remove the SDS. The resulting pellet was used for labeling with Alexa Fluor 647 NHS ester (succinimidylester, Invitrogen, Cat# A20006) in 100 mM sodium bicarbonate buffer pH 8.3. The dye was dissolved in 100% DMSO as per the manufacturer instructions at a concentration of 10 mg/mL. The labeling reaction (100 µL) was carried out for 2 h at room temperature with continuous agitation. After the incubation, excess dye was removed by washing 3–4 times in sodium bicarbonate buffer. The resulting pellet was stored in 4 °C and next day, one washing was done with MQ. The resulting pellet was digested with 1 mg/mL of purified $Ec$ BcsZ cellulase (endo-β-1,4-glucanase, GH-8) in 250 µL reaction volume at 37 °C for 4 h in 20 mM sodium phosphate buffer pH 7.2. Control reactions were performed by omitting BcsZ. The digested samples were centrifuged at 21,200 × $g$ for 20 min at room temperature and the soluble oligosaccharides were collected and dried using a centrifugal evaporator at low heat settings. The dried sample was resuspended in 15 µL urea and then analyzed by PACE as described previously[32]. Briefly, from each sample, 2.5 µL was loaded onto the 240 × 180 × 0.75 mm polyacrylamide gel comprising a stacking gel with 10% polyacrylamide and a resolving gel with 20% acrylamide, both containing 0.1 M Tris-borate pH 8.2. Gels were run in 0.1 M Tris-borate buffer in a Hoefer SE660 electrophoresis tank (Hoefer Inc, Holliston, MA, USA) at 200 V for 30 min and then 1000 V for 2 h before imaging using a G-Box Chemi-XX6 (Syngene, Cambridge, UK). Images were acquired with the GeneSys software (Syngene, version 1.8.5.0) based on the excitation and emission wavelength of the fluorophore (651/672 nm). A mixture of glucose and cello-oligosaccharides with DP 2–6 [each with 20 µM final concentration; $(Glc)_{1-6}$ ladder/Standard] was also loaded onto the gel as an internal mobility marker. For visualization, the marker was labeled with 8-aminonapthalene-1,3,6-trisulfonic acid (ANTS, Invitrogen, Cat# A350) as described previously[32]. The ANTS labeled standard was imaged using the same G-Box equipped with a long-wavelength (365 nm) UV tube, and short-pass detection filter (500–600 nm). Additional standard reactions were set up initially to design these experiments wherein pure $Ec$ pEtN cellulose (kindly provided by Lynette Cegelski, University of Stanford) or the unmodified phosphoric acid swollen cellulose was labeled with Alexa Fluor 647 NHS ester and digested with BcsZ.

## Western blotting

Following SDS-PAGE, the protein was transferred to a nitrocellulose membrane using a BioRad Transfer system. After blocking the membranes with 5% nonfat milk in Tris-buffered saline and 0.1% Tween 20 buffer (TBST), the membranes were washed with TBST and incubated with primary antibody at 4 °C overnight. Specific primary antibodies were used, including Penta-His (Qiagen, Cat# 34660; dilution 1:2000) for detecting BcsA chimera, $Rs$ BcsA and poly-histidine-tagged BcsG, and Anti-Flag M2 (Sigma, Cat#3165; dilution 1:10,000) for detection of FLAG-tagged BcsG. The membranes were then washed three times by incubating with fresh TBST for 10 min before incubation with anti-mouse IgG conjugated to a DyLight 800 fluorescent marker (Rockland, Cat# 610-145-002; dilution 1:5000) for 1 h at room temperature. The membranes were washed three times with fresh TBST and visualized using an Odyssey Light scanner at wavelengths 700 and 800 nm.

## Mass spectrometry

Protein identification of BcsA NTD by mass spectrometry was performed on Coomassie stained and excised SDS-PAGE gel band. For this purpose, purified $Ec$ NTD-BcsG complex from the gel filtration elution peak was loaded onto a 17.5% SDS-PAGE gel. After staining and destaining and rinsing in MQ, the desired band migrating close to the $Ec$ BcsA NTD molecular weight was excised and submitted to Biomolecular Analysis Facility at the University of Virginia.

The gel pieces were digested in 20 ng/µL trypsin in 50 mM ammonium bicarbonate on ice for 30 min. Any excess enzyme solution was removed and 20 µL 50 mM ammonium bicarbonate added. The sample was digested overnight at 37 °C and the released peptides were extracted from the polyacrylamide in a 100 µL aliquot of 50% acetonitrile/5% formic acid. The samples were purified using C18 tips. This extract was evaporated to 20 µL for MS analysis.

The liquid chromatography-mass spectrometry (LC-MS) system consisted of a Thermo Electron Orbitrap Exploris 480 mass spectrometer system with an Easy Spray ion source connected to a Thermo 75 µm × 15 cm C18 Easy Spray column. 5 µL of the extract was injected and the peptides eluted from the column by an acetonitrile/0.1 M formic acid gradient at a flow rate of 0.3 µL/min over 2 h. The nanospray ion source was operated at 1.9 kV. The digest was analysed using the rapid switching capability of the instrument acquiring a full scan mass spectrum to determine peptide molecular weights followed by product ion spectra (Top10 HCD) to determine amino acid sequence in sequential scans. This mode of analysis produces ~25,000 MS/MS spectra of ions ranging in abundance over several orders of magnitude. The data were analysed by database searching using the Sequest search algorithm against the $Ec$ BcsA protein and Uniprot $Ec$.

## Electron microscopy

**Grid preparation and data acquisition.** Cryo-EM grids for the nanodisc reconstituted sample or the detergent solubilized samples were prepared in a similar manner on Quantifoil R1.2/1.3 Cu 300 mesh or C-Flat 1.2/1.3–3 Cu grids respectively. Grid preparation was optimized for each protein individually in regard to the protein concentration such that cryo-EM grids was prepared with purified chimeric BcsAB-BcsG complex at a concentration of 1.6 mg/mL; $Ec$ Bcs complex with BcsB-BcsC fusion at a concentration of about 2.2 mg/mL, and BcsC reconstituted in nanodiscs at a concentration of about 2.5 mg/mL. To explore the interactions between the IMC and the outer membrane porin BcsC, $Ec$ Bcs complex reconstituted in nanodisc was incubated with the purified periplasmic domain of BcsC (TPR#1–18) at a molar ratio of 1:5 for 1 h on ice before cryo grid preparation. Likewise for the data collection with BcsZ, BcsC was incubated with the purified BcsZ at a molar 1:1.5. For data collection of BcsC nanodiscs incubated with cellotetraose, in addition to the 5 mM cellotetraose present during the BcsC nanodisc reconstitution mixture, an additional 10 mM cellotetraose was added to the final sample before grid preparation.

Grids were glow-discharged in the presence of amylamine and 2–2.5 µl sample was applied to each grid and blotted and plunge-frozen

in liquid ethane using a Vitrobot Mark IV plunge-freezing robot operated at 4 °C and 100% humidity. Blotting force and time was optimized individually for each sample: force of 4 and time 6 s for both the chimeric BcsAB-BcsG complex and the BcsB-BcsC fusion *Ec* complex; force 4 and blotting time 5 s for the *Ec* Bcs complex in nanodiscs; and force 4 and 4 s blotting time for BcsC. Grids were loaded into a Titan Krios electron microscope equipped with a K3/GIF detector (Gatan) at the Molecular Electron Microscopy Core (University of Virginia School of Medicine). Data were collected using EPU in counting mode at a magnification of 81 K, pixel size of 1.08 Å, and a total dose of 51 e$^-$/Å$^2$, with a target defocus varying from −1 to −2.4 μm.

**Data processing.** Data was processed using cryoSPARC v4.0.0[28] or Relion v3.1.3[50], following similar processing pipelines. Initially, the movies were imported and the images were first normalized by gain reference. Beam induced motion correction was performed using patch motion correction and contrast transfer function (CTF) parameters were estimated using patch CTF estimation. Micrographs were manually curated and those with outliers in defocus value, astigmatism, total full-frame motion, ice thickness, and low resolution (below 4.5 Å) were removed. From these high-quality micrographs, particles were selected using Blob picker, extracted, and used to generate 2D classes for template-based particle picking. After extraction of these particles and 2D classification, ab initio models were generated and subjected to several rounds of heterogeneous refinement. Selected particles and volumes were subjected to non-uniform refinement. For all datasets except BcsZ, this was followed by local refinement and further 3D classification using model or solvent based masks as described in the workflows. The best 3D class was subjected to local refinement again. Model building was performed in Coot starting with AlphaFold2-predicted models of the complex of BcsB and BcsC's N-terminal TPR#1–4 as well as for BcsC. The cryo-EM structure of the *Ec* BcsB hexamer (PDB: 7L2Z) and the crystal structure of BcsZ (PDB 3QXQ) were used as the corresponding initial models. The coordinates were rigid body docked into the corresponding volumes using Chimera. Models were manually refined in Coot (0.9.8.92)[51] or ISOLDE (1.6.0)[52] and real space refined in PHENIX (v1.21.1-5286-000)[53]. The model of BcsA in association with the BcsG trimer was predicted by AlphaFold2 and docked into the corresponding map based on BcsA's location. The individual BcsG subunits and BcsA's NTD were then rigid bodies docked into their densities. The model of the BcsA-BcsG3 complex containing the BcsB hexamer was obtained after rigid body refinement against a composite map generated from the best BcsB and BcsA-BcsG3 volumes. Figure representations were generated using Chimera (v1.17.1)[54], ChimeraX (v1.7.1)[55] and PyMOL (v3.0.2)[56].

#### Reporting summary

Further information on research design is available in the Nature Portfolio Reporting Summary linked to this article.

## Data availability

Coordinates of the BcsA-BcsB6_BcsG3 complex have been deposited at the PDB alongside the cryo-EM map with accession codes 9B8V and EMD-44359. The coordinates for the BcsB-BcsC complex, BcsC in the absence and the presence of cellotetrose, and the BcsZ tetramer have been deposited at the Protein Data Bank with accession codes 9B8I and EMD-44346, 9B8A and EMD-44336 and 9B8H and EMD-44345, and 9B87 and EMD-44334, respectively. Previously published structures used in this study are available in the Protein Data Bank under accession codes 5FGN, 6PD0, 4P00, 7L2Z, 3QXF and 3QXQ. AlphaFold predicted structures of full-length *E. coli* BcsG AF-P37659-F1 and BcsC AF-P37650-F1 used in this study are publicly available. Source data are provided in this paper.

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

## Acknowledgements

We are grateful to Michael Purdy and Kelly Dryden of the Macromolecular Electron Microscopy Facility at U.V.A. for support during cryo-EM data collection and Louis Wilson for help with PACE analysis. We thank Phillip Stansfeld and Robin Corey for insightful discussions. The work was supported by NIH grant R35GM144130 awarded to J.Z. L.C. acknowledges support from NSF grant award 2001189. S.A.C. acknowledges support from the NIH postdoctoral fellowship award NIHF32GM149117. J.Z. is an investigator of the Howard Hughes Medical Institute. This article is subject to HHMI's Open Access to Publications policy. HHMI lab heads have previously granted a nonexclusive C.C. BY 4.0 license to the public and a sublicensable license to HHMI in their research articles. Pursuant to those licenses, the author-accepted manuscript of this article can be made freely available under a C.C. BY 4.0 license immediately upon publication.

## Author contributions

P.V. performed all cloning and protein purification procedures, performed all biochemical assays, in vivo pEtN cellulose isolation, and processed all new cryo-EM data. P.V. and R.H. performed grid preparation and cryo-EM data collection. J.Z. processed the BcsG cryo-EM data. S.A.C. developed the periplasmic pEtN cellulose isolation protocol and performed the ssNMR measurements and analysis. L.C. assisted in data analysis and preparation of the manuscript. J.Z. generated the initial manuscript. All authors interpreted the data and revised the manuscript.

## Competing interests

The authors declare no competing interests.
