## [Peer Review File · Nature Communications]

Insights into phosphoethanolamine cellulose synthesis and secretion across the Gram-negative cell envelopeREVIEWER COMMENTS

Reviewer #1 (Remarks to the Author):

Summary Comments to the authors:

The work by Verma et al. describes for the first time how BcsG fits into the cellulose synthase complex of *E. coli*, and also substantially advances our understanding of polymer passage from BcsA through to BcsC and release by BcsZ. The structures of the ABGC complex reveal important insight into the orientation and stoichiometry of the individual Bcs subunits that match biological relevance. This work required structural analysis by a number of means (cryo-EM, AlphaFold2 and crystal structure data manipulation) on all of the proteins and across different species (hybrid formation with Rs enzymes) and is a remarkable feat, as it delineates for the first time important residues in the binding surface (which previously had been a mystery) and uncovers a pathway for the export of cellulose.

Revision suggestions:

Main Comment:

The manuscript was very well-written and laid out. The BcsG and BcsC data is solid and analysis of the results is appropriate. I had very few comments with respect to these proteins (other than figure representation/edits). However, I found that I got caught up in the BcsZ data and I am not sure about the mislocalized idea for BcsZ. Can we exclude the idea that perhaps BcsZ is present for the release of the polymer from the cell only. The off-the-shelf cellulases would do much the same by cutting it into smaller (more manageable chunks by the cell). It is important to remember that there is no energy put into this system beyond the successive addition of UDP-glucose from within the cell. Perhaps the activity of the cellulase periodically relieves the constraints of continuing to push extremely long polymers out of the cell? This prevent the system from “stalling” rather than mislocalization. In the absence of BcsZ, is there a substantial increase of polymer in the

periplasm that would support mislocalization over stalling? I know that is the case for the BcsC mutant, but I am not sure it is the same for BcsZ. Some rewording would be needed to account for these scenarios if this manuscript moves forward. This would help future scientists that may try to decipher these possibilities rather than overcoming dogma that focuses on one or the other.

Other Specific Minor Comments:

L328– Speculative that this would even take place -“bending” - reword

L362 - Did you test for a buildup of cellulose in the periplasm? This would also help answer the structural necessity of BcsZ aside from the activity.

L394 - Can you comment more on the biological relevance of this?

L429 – Can you comment more on the biological relevance of this?

L457 – What was the dp of the cellulose made by the off the shelf cellulases? Presumably this would change with the more/less active an enzyme is. Were they able to form biofilms and were they different from WT-BcsZ containing biofilms?

Figure 1

- Is the arrow on the D panel backwards? Doesn't the molecule swing up instead of down between the two panels?

- Can you make it clear with your wording that just the membrane portion of the BcsB subunits is shown in e)?

- Spelling mistake before e)

- Label BcsB in e) top panel so it is easier to follow when flipped to the second panel

-F and D panels seem to be squashed together.

-For panel F, The Strep-tactin lanes seem to be from more than one gel and have been squished together to look like one. I think this is a bit of a misrepresentation that needs to be corrected so that they do not mislead the reader.

Figure 2

- Is the arrow in panel A wrong? It looks like a top-down view and not bottom up as the arrow suggests.
- In the F panel, please include what C1-8 are referring to for the reader.

Figure 3

- Is the arrow in panel A wrong? It looks like a top-down view and not bottom up as the arrow suggests.

Figure 6

- modify to include “stalling” rather than mislocalization as a possibility.

Figure S4

- What does R/E_A stand for in panel G? Please add this in brackets in the description after chimera BcsA.

Reviewer #2 (Remarks to the Author):

In this work, Verma and colleagues study the pEtN cellulose formation and translocation mechanism in Enterobacteriaceae. The authors demonstrate that the catalytic subunit BcsA recruits three copies of BcsG, a membrane-anchored periplasmic pEtN transferase, via its N- and C-terminal domains. The authors show that the BcsC subunit binds the sixth subunit of the BcsB semicircle via its extreme N-terminus to establish an envelope-

spanning Bcs complex. Interestingly, BcsC binds a putative cello-oligosaccharide at its TPR, likely facilitating translocation across the periplasm.

Based on solid structural/biochemical/biological data a molecular mechanism/model for pEtN cellulose formation and translocation is proposed. The manuscript is very well written and, because of its novelty, it is therefore suitable for publication in Nature Communications. Please, find below two suggestions for improvement:

1. Please introduce the chemical reaction catalyzed by the pEtN transferase in the context of cellulose modification, as a Figure 1a.

2. I suggest improving the model presented in Figure 6. I think it is a good opportunity to generate/show a model entirely built on secondary structure elements/surface representation of the different subunits, highlighting the location of cellulose, along its synthesis and transport processes. Several orientations will be necessary, but it would be an important exercise to clarify the mechanism/s, which will undoubtedly enrich the manuscript.

Reviewer #3 (Remarks to the Author):

This is a follow up study to previously published studies on bacterial cellulose synthase. The authors analyzed the EM data with greater resolution to determine that: 1. there are 3 BcsG subunits in the cellulose complex, 2. Purified cellulose synthase complex with BscG can add pEtN to cellulose, 3. BscB interacts with BscC, 4. BscC bound to cellulose, and 5. BscZ cellulase activity is needed to promote cellulose export. Together, the authors advance a model for cellulose modification and secretion.

Major issues:

1. Figure 1b – Can all the different subunits be labeled? Can a gray bar be placed in this panel to indicate the relative position of the inner membrane?

Figure 1d – can the periplasmic domain for BscG be fitted to this model? Or is this density missing? If missing, does this mean BscG is cleaved?

2. Figure S2 – does the three classes indicate differences in number of BscG subunits, orientation of the BscG subunits, or something else? Is there actually heterogeneity in the complexes or are they different states of the same complex? Can this be added to the discussion?

3. Figure 2a and 2e - Can a gray bar be placed in this panel to indicate the relative position of the membrane? Figure 2f – what is the signal for pure cellulose without pEtN modification?

4. Figure S3 – Can the regions being replaced in the *R. sphaeroides* with *E. coli* be indicated schematically?

5. Figure 3a - Can a gray bar be placed in this panel to indicate the relative position of the membrane? I can't figure out where this interaction is occurring relative to the rest of the complex.

6. Figure 4 – Can discussion be added as to why there are gaps in the periplasm for cellulose to escape? Seems like the protein can be folded such that there are no gaps for escape into the periplasm. Is this necessary feature of export of polysaccharides or is this something that allow for additional modification of the polymer during transit out of the cell?

7. Figure 5d, 5e, and 5f – unclear the meaning of the BscZ tetramer? Is this the state that is found in the periplasm? If this is not observed in cells, perhaps these panels and companion text should be removed from this manuscript.

8. Lines 438-447 – does the escaping polymer into the periplasm cause problems? There might be some analogy to alginate secretion and alginate lyase. In the absence of AlgL, *Pseudomonas aeruginosa* accumulate alginate in the periplasm and explodes (PMID: 16177314). This would be an interesting comparison to discuss. Is cellulose accumulation in the periplasm ever observed in the *bscZ* mutant? If not, why not?

Minor issues:

1. Mutation into the NTD or other critical region of BscA chimera could help to support the claim that these residues are important for BscG interaction and activity.

2. Does other lipid head groups compete with PE lipid for the pEtN transfer site in BscG? Could this introduce modification to cellulose?

Reviewer #4 (Remarks to the Author):

We would like to thank the reviewers for taking the time to review our manuscript and providing constructive criticism. We have addressed all points raised by the reviewers in the manuscript and the responses below.

Reviewer #1 (Remarks to the Author):

Main Comment:

The manuscript was very well-written and laid out. The BcsG and BcsC data is solid and analysis of the results is appropriate. I had very few comments with respect to these proteins (other than figure representation/edits). However, I found that I got caught up in the BcsZ data and I am not sure about the mislocalized idea for BcsZ. Can we exclude the idea that perhaps BcsZ is present for the release of the polymer from the cell only. The off-the-shelf cellulases would do much the same by cutting it into smaller (more manageable chunks by the cell). It is important to remember that there is no energy put into this system beyond the successive addition of UDP-glucose from within the cell. Perhaps the activity of the cellulase periodically relieves the constraints of continuing to push extremely long polymers out of the cell? This prevent the system from “stalling” rather than mislocalization. In the absence of BcsZ, is there a substantial increase of polymer in the periplasm that would support mislocalization over stalling? I know that is the case for the BcsC mutant, but I am not sure it is the same for BcsZ. Some rewording would be needed to account for these scenarios if this manuscript moves forward. This would help future scientists that may try to decipher these possibilities rather than overcoming dogma that focuses on one or the other.

> We currently do not know how much cellulose accumulates in the periplasm in the absence of BcsZ or the presence of inactive variants. The idea that the enzyme may also be involved in restarting stalled biosynthetic complexes is interesting and is now discussed. However, if stalling occurs due to cellulose aggregation on the cell surface, one might expect less dramatic effects on Congo red staining of macrocolonies, since a significant portion of cellulose would be surface exposed. Further, because the only energy input occurs at the inner membrane, it is unclear how a cleaved polymer would clear the secretion system as no additional translocation force can be exerted on a cleaved polymer. In this case, the polymer would have to slide through BcsC based on diffusion. This seems unlikely for a water-insoluble polymer. Nevertheless, we rephrased the discussion to include stalling relief as a potential function of BcsZ

An additional (untested) function of BcsZ may be in quality control. If the enzyme is unable to degrade pEtN modified cellulose, its function could also be to remove unmodified stretches from the translocating polymer to maximize the deposition of pEtN cellulose on the surface, thereby promoting biofilm adhesion properties.

Other Specific Minor Comments:

L328– Speculative that this would even take place –“bending” – reword

- We replaced bending with ‘tilting’. Contrary to general assumptions, cellulose is quite flexible and easily adopts bent or curved conformations.

L362 - Did you test for a buildup of cellulose in the periplasm? This would also help answer the structural necessity of BcsZ aside from the activity.

- At present, we do not have suitable probes that would allow us to locate cellulose in the periplasm, presumably via gold labeling in thin section TEM. We are working on these detection methods. Currently, we do not know whether significant amounts of cellulose accumulate in the periplasm upon BcsZ inactivation or whether cellulose biosynthesis stalls early on due to cellulose aggregating in the periplasm.

L394 - Can you comment more on the biological relevance of this?

- We are pointing this out to stress that the BcsZ subunits of the tetramer likely function independently, most likely on different cellulose polymers. The arrangement of the catalytic pockets and the directionality with which they bind cellulose prevent a BcsZ tetramer from interacting with and cleaving a contiguous cellulose polymer running diagonally across the tetramer interface. Nevertheless, the ability of a tetramer to interact with mislocalized cellulose in multiple directions, enabled by the tetramer, may increase its catalytic efficiency.

L429 – Can you comment more on the biological relevance of this?

- Our grafting experiment demonstrates that the association of BcsA's NTD with BcsG's TM region can be used to recruit cellulose modifying enzymes to the biosynthesis site. If transferrable to cellulose synthases from other systems, this could be employed to generate novel biomaterials.

L457 – What was the dp of the cellulose made by the off the shelf cellulases? Presumably this would change with the more/less active an enzyme is. Were they able to form biofilms and were they different from WT-BcsZ containing biofilms?

> The degree of polymerization (dp) of cellulose produced in the presence of Cel9M (an off-shelf cellulase) or in the presence of BcsZ appears to be high. Predicting the dp via ¹³C solid-state NMR is feasible for low or moderate dp celluloses, but challenging for higher dp celluloses due to low signal intensities unless the samples are hydrolyzed (Schmidt-Rohr et al, 2022). We have now included a ¹³C NMR spectrum of the cellulosic material obtained in the presence of Cel9M as Fig. S10b. It overlays nearly perfectly in the 80-110 ppm region thus indicating similar dp. There is a small reduction in the C8 carbon intensity and thus concomitant changes in the relative C6 and C7 carbon intensities which differ for modified and unmodified cellulose indicating a lower level of pEtN modification in this construct.

Figure 1

- Is the arrow on the D panel backwards? Doesn't the molecule swing up instead of down between the two panels?
- Can you make it clear with your wording that just the membrane portion of the BcsB subunits is shown in e)?
- Spelling mistake before e)

- Label BcsB in e) top panel so it is easier to follow when flipped to the second panel
- F and D panels seem to be squashed together.
- For panel F, The Strep-tactin lanes seem to be from more than one gel and have been squashed together to look like one. I think this is a bit of a misrepresentation that needs to be corrected so that they do not mislead the reader.

➤ Figure 1 has been substantially revised to address these excellent suggestions. Thank you.

Figure 2

- Is the arrow in panel A wrong? It looks like a top-down view and not bottom up as the arrow suggests.
- In the F panel, please include what C1-8 are referring to for the reader.

> These points have been addressed.

Figure 3

- Is the arrow in panel A wrong? It looks like a top-down view and not bottom up as the arrow suggests.

> The arrow has been revised to avoid confusion.

Figure 6

- modify to include “stalling” rather than mislocalization as a possibility.

➤ Done as suggested

Figure S4

- What does R/E_A stand for in panel G? Please add this in brackets in the description after chimera BcsA.

> R/E refers to the Rhodobacter (R)/E. coli (E) BcsA chimera. This is now explained in the caption.

Reviewer #2 (Remarks to the Author):

1. Please introduce the chemical reaction catalyzed by the pEtN transferase in the context of cellulose modification, as a Figure 1a.

> Included as requested.

2. I suggest improving the model presented in Figure 6. I think it is a good opportunity to generate/show a model entirely built on secondary structure elements/surface representation of the different subunits, highlighting the location of cellulose, along its synthesis and transport processes. Several orientations will be necessary, but it would be an important exercise to clarify the mechanism/s, which will undoubtedly enrich the manuscript. (I tried and proposed fig is in box, but can be refined more if you like the idea)

> We generated such a ‘pseudo atomic’ model, which is now presented in Figure S11. However, we have several concerns regarding this model because (a) the AlphaFold predicted structure of

BcsC had to be arbitrarily ‘straightened’ to remove a highly unlikely kink in BcsC’s TPR solenoid, and (b), at least one catalytic domain of the BcsG trimer clashes with the BcsB crown (based on the AlphaFold predicted full length structure). These caveats are stated in the corresponding caption.

Reviewer #3 (Remarks to the Author):

Major issues:

1. Figure 1b – Can all the different subunits be labeled? Can a gray bar be placed in this panel to indicate the relative position of the inner membrane?

>> Panel 1b has been revised. The BcsB hexamer and detergent micelle regions are now indicated.

Figure 1d – can the periplasmic domain for BcsG be fitted to this model? Or is this density missing? If missing, does this mean BcsG is cleaved?

> We explain in the text that only BcsG’s transmembrane region is resolved in the cryo-EM map, although the full-length protein is present in the analyzed sample. However, at low contour levels, discontinuous density is observed within the open portion of the BcsB semicircle (Fig. 3a and Acheson et al., NSMB 2021); yet, at best, this density only represents a fraction of the volume occupied by the BcsG catalytic domains.

Based on the reviewer’s request, we have tried to dock the full-length AlphaFold predicted model of BcsG into the *Ec* complex based on the observed cryo-EM density for the three copies of BcsG TM segments (Fig S11). The clashes between the first copy of BcsG catalytic domain and first copy of BcsB associated with BcsA are evident. This is of functional importance as the catalytic domain has to be flexible in order to receive and transfer a lipid-derived pEtN group.

2. Figure S2 – does the three classes indicate differences in number of BcsG subunits, orientation of the BcsG subunits, or something else? Is there actually heterogeneity in the complexes or are they different states of the same complex? Can this be added to the discussion?

> This is a routine classification in cryo EM data processing and does not reflect on the numbers of BcsG subunits observed. Rather, it separates particles into classes that can be refined to higher resolution, and those that are too disordered and of insufficient quality. We certainly observe heterogeneity of the Bcs complex regarding the presence of the cytosolic subunits (BcsQ/R/E) as well as the number of BcsB subunits forming the semicircle. This heterogeneity most likely results from complex dissociation during purification. Hence, we hesitate to draw any functional conclusions. We have not observed any variability in terms of number of BcsG subunits present in the complex. This seems to be always three subunits.

3. Figure 2a and 2e – Can a gray bar be placed in this panel to indicate the relative position of the membrane?

> Done as requested for panel 2a. Detergent micelle region has been shown for panel 2e for clarity.

Figure 2f – what is the signal for pure cellulose without pEtN modification?

> We have included this comparison as Fig S6a. A direct comparison of unmodified and pEtN modified cellulose is also provided by Thongsomboon, W., et al, Science 2018.

4. Figure S3 – Can the regions being replaced in the *R. sphaeroides* with *E. coli* be indicated schematically?

> Done

5. Figure 3a - Can a gray bar be placed in this panel to indicate the relative position of the membrane? I can't figure out where this interaction is occurring relative to the rest of the complex.

> We indicated the cytosolic and periplasmic sides in panel 3a to orient the reader. Also, we hope that the final model presented in Fig. 6 helps with the overall orientation.

6. Figure 4 – Can discussion be added as to why there are gaps in the periplasm for cellulose to escape? Seems like the protein can be folded such that there are no gaps for escape into the periplasm. Is this necessary feature of export of polysaccharides or is this something that allow for additional modification of the polymer during transit out of the cell?

> We assume that the reviewer refers to the openings of the TPR solenoid that could allow the cellulose polymer to escape into the periplasm. While we don't have structural insights into BcsC bound to a cellulose polymer sufficiently long to span the entire solenoid, it seems unlikely that the TPRs would rearrange to form a sealed channel across the periplasm. The BcsC architecture appears to be designed so that the concave sides of its solenoid aid in ligand binding. In some related systems, the translocating polymer is indeed modified in the periplasm, hence accessibility is necessary.

7. Figure 5d, 5e, and 5f – unclear the meaning of the BcsZ tetramer? Is this the state that is found in the periplasm? If this is not observed in cells, perhaps these panels and companion text should be removed from this manuscript.

> We do not know what the oligomeric state of BcsZ is in the periplasm. Here, we report the cryo-EM structure of the BcsZ tetramer. This tetramer exists under mild buffer conditions (in contrast to crystallization conditions that revealed the tetramer earlier); hence it is likely that the tetramer also forms in the periplasm. We are currently addressing the localization and distribution of BcsZ *in vivo* using super-resolution fluorescent microscopy approaches.

8. Lines 438-447 – does the escaping polymer into the periplasm cause problems? There might be some analogy to alginate secretion and alginate lyase. In the absence of AlgL, *Pseudomonas aeruginosa* accumulate alginate in the periplasm and explodes (PMID: 16177314). This would be an interesting comparison to discuss. Is cellulose accumulation in the periplasm ever observed in the *bscZ* mutant? If not, why not?

> Thank you for pointing this out and apologies for the oversight. It is indeed possible that cellulose accumulation in the periplasm causes cell stress and/or lysis, although alginate may have a more severe impact on osmolarity, compared to cellulose. Currently, we have no means to unambiguously localize cellulose in the periplasm. Our attempts to isolate pEtN cellulose from systems lacking the outer membrane porin BcsC were unsuccessful, suggesting that cellulose biosynthesis stalls if the polymer cannot be secreted. We are currently developing specific probes that can be used for nanogold labeling of cellulose, for example for thin section TEM. Our observation is that co-expression of a BcsZ with the other Bcs components increases the overall expression yield, which could correlate with cell stress in the absence of a functional cellulase. The AlgL data is now discussed and referenced.

Minor issues:

1. Mutation into the NTD or other critical region of BscA chimera could help to support the claim that these residues are important for BscG interaction and activity.

> Pull down experiments of BcsG either with the R/E_A chimera (*Rhodobacter* BcsA fused with the NTD of *E. coli* BcsA) or the isolated *Ec* BcsA_NTD domain show that the NTD is sufficient to recruit the BcsG trimer, hence we don't think that additional mutagenesis is necessary. Biochemical analysis of BcsG's catalytic domain alone demonstrate that it is active with soluble substrates (not lipids) in the absence of the transmembrane region. Hence, we don't expect any stimulatory role of BcsA's NTD on catalytic activity of BcsG. The main purpose of the NTD domain is most likely to place BcsG in close proximity to the translocating cellulose polymer.

2. Does other lipid head groups compete with PE lipid for the pEtN transfer site in BscG? Could this introduce modification to cellulose?

> This is a very interesting thought and would likely require engineering of BcsG. This is certainly on our to-do list.

Reviewer #4 (Remarks to the Author):

REVIEWERS' COMMENTS

Reviewer #1 (Remarks to the Author):

I appreciate the time and effort that has gone into addressing all of the reviewer's questions from the first submission. I feel that the manuscript is greatly improved and have only a few further minor comments listed below:

L439 – “grafting experiments”. The rebuttal description is clear with the meaning of this statement. Could the authors incorporate some of this into the body of the manuscript.

L379-382 – including the additional NMR is helpful. However, how this relates to differences in biofilm formation is still not addressed. Perhaps this can be examined in future studies.

Fig. 6 – Can BcsZ be represented in a more 3 dimensional way to match the “look” of all the other components?

Reviewer #2 (Remarks to the Author):

The authors answered all my comments. Congratulations on this really nice work.

Reviewer #3 (Remarks to the Author):

Thanks for addressing the issues raised.

Reviewer #4 (Remarks to the Author):

Reviewer #1 (Remarks to the Author):

I appreciate the time and effort that has gone into addressing all of the reviewer's questions from the first submission. I feel that the manuscript is greatly improved and have only a few further minor comments listed below:

L439 – “grafting experiments”. The rebuttal description is clear with the meaning of this statement. Could the authors incorporate some of this into the body of the manuscript.

- This has been revised as suggested.

L379-382 – including the additional NMR is helpful. However, how this relates to differences in biofilm formation is still not addressed. Perhaps this can be examined in future studies.

- As suggested by the reviewer, this has to await future analyses of pEtN cellulose engineered biofilms.

Fig. 6 – Can BcsZ be represented in a more 3 dimensional way to match the “look” of all the other components?

- Done as requested.

Reviewer #2 (Remarks to the Author):

The authors answered all my comments. Congratulations on this really nice work.

- Thank you

Reviewer #3 (Remarks to the Author):

Thanks for addressing the issues raised.

- Thank you

Reviewer #4 (Remarks to the Author):

- Thank you for your help.